# Human granulocytotropic anaplasmosis—A systematic review and analysis of the literature

**Sophie Schudel**[1,2], **Larissa Gygax**[1,2], **Christian Kositz**[1,3]\*, **Esther Kuenzli**[1,2], **Andreas Neumayr**[1,2,4]

**1** Swiss Tropical and Public Health Institute, Basel, Switzerland, **2** University of Basel, Basel, Switzerland, **3** Clinical Research Department, Faculty of Infectious and Tropical Diseases, London School of Hygiene & Tropical Medicine, London, United Kingdom, **4** Department of Public Health and Tropical Medicine, College of Public Health, Medical and Veterinary Sciences, James Cook University, Queensland, Australia

\* christian.kositz@lshtm.ac.uk

**Data Availability Statement:** The authors confirm that all data underlying the findings are fully available without restriction. All relevant data are

## Abstract

Human granulocytotropic anaplasmosis (HGA) is a zoonotic tick-borne bacterial infection caused by *Anaplasma phagocytophilum*. While most cases are reported from North America, HGA has been recognized as an emerging disease in several regions of the world in recent decades. Most available data comes from case reports, case series and retrospective studies, while prospective studies and clinical trials are largely lacking. To obtain a clearer picture of the currently known epidemiologic distribution, clinical and paraclinical presentation, diagnostic aspects, complications, therapeutic aspects, and outcomes of HGA, we systematically reviewed the literature and analyzed and summarized the data.

Cases of HGA are reported from all continents except from Antarctica. HGA primarily presents as an unspecific febrile illness (88.5% of the cases) often accompanied by thrombocytopenia (71.8% of the cases), abnormal liver injury tests (66.7% of the cases), and leukopenia (49.8% of the cases). Although we found complications reported in a total of 40.5% of the reviewed cases and severe and even life-threatening complications are not infrequent (e.g. acute renal failure 9.8%, multi organ failure 7.5%, ARDS 6.3%, a.o.), sequelae are rare (2.1% of the cases) and lethality is low (3.0% of the cases). Treatment with doxycycline shows a rapid response, with the fever subsiding in the majority of patients within one day of starting treatment. Unlike in human monocytotropic ehrlichiosis (HME), reports of opportunistic infections complicating HGA are rare. HGA during pregnancy does not appear to be associated with unfavorable outcomes. In addition, our analysis provides some evidence that HGA may differ in clinical aspects and laboratory characteristics in different regions of the world. Overall, the data analyzed indicates a non-negligible bias in reporting/publication, so a certain degree of caution is required when generalizing the data.

within the paper and its Supporting information files.

**Funding:** The author(s) received no specific funding for this work.

**Competing interests:** The authors have declared that no competing interests exist.

## Author summary

Human granulocytotropic anaplasmosis (HGA) is a bacterial disease caused by the bacterium *Anaplasma phagocytophilum* which is transmitted by tick bites. While often asymptomatic, symptomatic disease usually presents with non-specific symptoms such as fever, exhaustion, headaches, and muscle aches. Response to antibiotic treatment is usually fast and effective, and although complications may arise, most patients experience a favorable outcome. Most cases of HGA are reported from North America, but since the first description of HGA in 1994, cases of HGA have been reported worldwide. Since most available data on HGA comes from case reports, case series and retrospective studies, data on several aspects of the disease remains patchy. To obtain a better overview on various aspect of HGA we systematically reviewed the existing literature and compiled and analyzed the reported data on epidemiology, clinical presentation, complications, diagnosis, treatment, and outcome of HGA. By compiling and comparing data from different sources, we aimed to gain a more comprehensive understanding of HGA.

## Introduction

Anaplasmoses are tick-borne zoonotic infections caused by gram-negative, obligate intracellular bacteria of the family Anaplasmataceae in the order Rickettsiales. All members of the Anaplasmataceae have in common that they survive in vacuoles of host cells, which usually originate from bone marrow, but occasionally also from endothelial cells. The different Anaplasmataceae vary with regard to their tropism for certain cell lines. *Ehrlichia chaffeensis*, *E. canis* and *E. muris* e.g. infect mostly the monocytes and macrophages while *Anaplasma phagocytophilum* and *E. ewingii* infect mostly the granulocytes of their mammalian hosts. This tropism for certain cell lines is reflected in some of the historical names of the illnesses these pathogens were observed to cause in their natural animal as well as in the accidental human hosts. In the context of human infection, this is *human monocytotropic ehrlichiosis* (HME) and *human granulocytotropic anaplasmosis* (HGA).

### Human anaplasmosis

The taxonomic order of the Anaplasmataceae underwent significant changes over time, with the most relevant one being the identification of *Anaplasma* as a separate genus and its separation from the genus *Ehrlichia* in 2001 [1]. With this change of taxonomic order, the previously called human granulocytotropic ehrlichiosis (HGE, first described as a human pathogen presenting similarly to HME in 1994 in the United States [2]) became human granulocytotropic anaplasmosis (HGA). The best epidemiological data on HGA is available from the United States, where HGA is a notifiable disease (until 2008 reported combined with HME, since 2009 reported separately). Over time, a total of 51'999 cases of HGA were reported in the USA between 2000 and 2021. With some fluctuations (e.g. in 2020 due to the COVID-19 pandemic) the reported numbers steadily increased, reaching an all time high in 2021 [3]. For the period 2008–2012 a calculated incidence rate of 7.27 cases per million population was reported, which corresponded to a fivefold increase compared to the period 2000–2007 [4,5]. Today, HGA is the second most commonly reported tick-borne disease after Lyme disease in the United States [6].

HGA is distributed worldwide, especially in northern latitudes of North America, Europe, and Asia, where *Ixodid* ticks, *Haemaphysalis longicornis* ticks and *Haemaphysalis concinna* ticks (China), and multiple small mammals serve as vectors and reservoir. The other known

**Table 1. The currently known human pathogenic *Anaplasma* species and their known mammalian hosts and tick vectors.**

| *Anaplasma* spp. | Mammalian hosts | Major target cell / cell tropism | Reported tick vectors |
|---|---|---|---|
| *Anaplasma phagocytophilum* | Mice, rats, voles, horses, dogs, cats, sheep, cattle, deer, humans | Granulocytes | *Ixodes persulcatus, Ixodes scapularis, Ixodes ricinus, Haemaphysalis longicornis, Haemaphysalis concinna* |
| *Anaplasma platys* | Dogs, humans | Platelets | *Rhipicephalus sanguineus* |
| *Anaplasma capra* | Wild and domestic ruminants, humans | Unknown | *Ixodes persulcatus* |
| *Anaplasma bovis* | Wild and domestic ruminants, (humans*) | Monocytes, erythrocytes# | *Rhipicephalus* spp., *Amblyomma* spp., *Haemaphysalis longicornis* |
| *Anaplasma ovis* | Sheep, goats, wild ruminants, (humans*) | Erythrocytes# | *Dermacentor* spp., *Rhipicephalus spp.*, *Haemaphysalis longicornis* |
| *Candidatus* Anaplasma sparouinense | ?, (humans*) | Erythrocytes | ? |

\* to date, only very few isolated human infections have been reported.

# cell tropism found in the natural animal host but unclear in human infection.

*Anaplasma* species are primarily zoonotic pathogens (Table 1) although a few isolated human infections due to *A. bovis*, *A. ovis*, *A. platys*, *and A. capra* have been reported [7–12]. Recently, a novel human pathogenic *Anaplasma* species, *Candidatus* A. saprouinense, was described in a splenectomized patient in French Guiana, of which the animal reservoir and the transmitting vector remains to be identified [13].

Besides infections transmitted by ticks, cases of HGA acquired by blood-transfusion have been reported.

## Clinical presentation

Human anaplasmosis presents as a febrile illness manifesting one to two weeks after the bite of an infected tick. The fever is accompanied by unspecific symptoms like weakness, malaise, headache, myalgia, arthralgia, nausea, and vomiting and rarely a rash is present. Most infections are uncomplicated and self-limiting (and most likely even asymptomatic), but severe complications, including septic shock, acute respiratory failure, renal failure, and multi-organ failure may occur and even be fatal. The higher incidence and risk for life-threatening complications in older patients suggests that host factors are important parameters influencing disease severity.

Although human infection caused by all members of the family Anaplasmataceae have been, and often continue to be, generically referred to as «ehrlichiosis», it is increasingly apparent that the clinical manifestations and causative agents are distinct [14].

## Diagnostic

Diagnosis of HGA rests primarily on clinical suspicion, epidemiological plausibility, and suggestive unspecific routine blood laboratory results, as the availability of sensitive and specific acute diagnostic tests (i.e. PCR) is often limited. Since the prognosis worsens and the risk for complications increases if treatment is delayed, it is recommended to start empirical antimicrobial treatment immediately if there is clinical suspicion and regardless of diagnostic tests [14]. Routine blood laboratory investigations often show cytopenia, particularly leukopenia and thrombocytopenia, elevated liver injury test levels (transaminases, alkaline phosphatase), and elevated C-reactive protein levels [15]. Light microscopic examination of blood smears (stained with e.g. Wright's, Diff-Quik, Giemsa) may be diagnostic if the typical, intracellularly located inclusions called *morulae* (from the Latin word for *mulberry*) are detectable. These

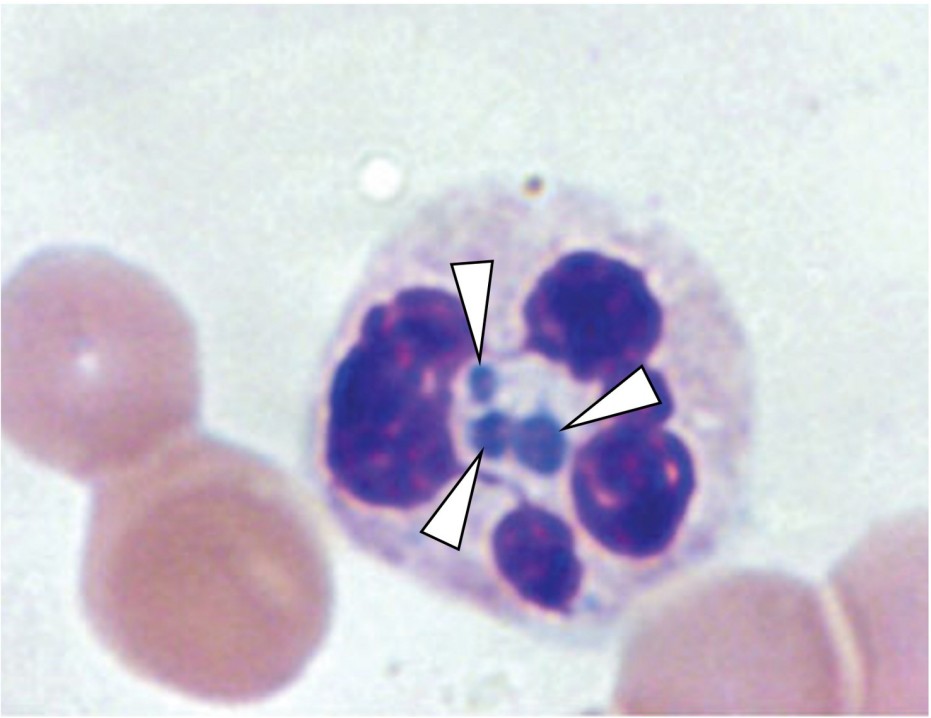

**Fig 1. Microscopic detection of *morulae* in blood films pathognomonic for pathogens from the family Anaplasmataceae.** Light microscopy of peripheral blood smear preparations (magnification ×1'000): granulocyte with intracellular *Anaplasma phagocytophilum* morulae (white arrows) surrounded by the granulocyte's segmented nucleus. The Figure was modified from [16].

*morulae* represent clustered microcolonies of the pathogen within the host cell vacuole (Fig 1). Although this method is rapid, it is relatively insensitive compared to other confirmatory tests, especially beyond the acute phase (first week of infection) with *morulae* being detectable in 25–75% of the blood smear examinations in HGA [14,15]. Besides microscopy, polymerase chain reaction (PCR) assays, serology, immunostaining of biopsy/autopsy material, and cell culture are used to diagnose HGA. Table 2 gives an overview on the advantages, disadvantages and applications of the different diagnostic methods available.

## Treatment and outcome

The treatment of choice for HGA is doxycycline or tetracycline and in most cases clinical response is rapid. The observation that mildly affected patients recover spontaneously even without specific therapy, as well as the high seroprevalence rates reported in some populations, suggest that most infections are mild, self-limiting and to a large extent probably subclinical [1,15].

National surveillance data of HGA from the United States shows a hospitalization and overall case fatality rate of 31% and 0.3%. The case fatality rate is highest in the elderly ≥70 years, but still less than 1% [19].

With our systematic review of HGA presented here, we aim to provide clinicians with a comprehensive summary of the available data, focusing on the clinically relevant core aspects (Note that an analogously compiled systematic review of human monocytotrophic ehrlichiosis (HME) was conducted in parallel by our group).

**Table 2. Overview of laboratory methods available to diagnose HME and HGA and their advantages, disadvantages and application/use [14,15,17,18].**

| Diagnostic method | Advantages | Disadvantages | Application/use |
|---|---|---|---|
| Microscopy of blood smear or buffy coat preparation | Widely available | Limited sensitivity (demands expertise, depends on density of *morulae*); does not allow reliable species differentiation; highest sensitivity during acute phase/first week of infection; limited specificity (does not allow conclusive species differentiation) | Used in the acute phase of infection when PCR is not available |
| PCR | High specificity (allows species differentiation); high sensitivity in the acute phase of infection; also suitable for biopsy/ autopsy samples | Decreased sensitivity beyond the acute phase/first week of infection and after administration of appropriate antibiotics | Used for diagnosis in the acute phase of infection; availability often limited to larger/reference laboratories |
| Serology (IFA, ELISA) | Enables retrospective diagnosis beyond the acute phase of infection | Not useful in acute phase of infection (due to delayed seroconversion); confirmation of diagnosis demands paired samples (acute and convalescent serum); limited specificity due to persisting antibodies after infection and cross-reactivity of assays with other Anaplasmataceae; possibly decreased sensitivity after early administration of appropriate antibiotics | Paired serology by IFA is the serological gold standard but the result will only be retrospectively available; used for epidemiological studies |
| Immunostaining of biopsy / autopsy tissue | High specificity; can also be applied to biopsy / autopsy samples | Availability limited to larger/reference laboratories; decreased sensitivity after administration of appropriate antibiotics | Useful for confirming the diagnosis in fatal cases where diagnostic levels of antibodies did not develop before death |
| Cell culture | High specificity | Low sensitivity; time and resource demanding; decreased sensitivity after administration of appropriate antibiotics | Diagnostic reference standard, but availability largely restricted to reference and research laboratories |

IFA, immunofluorescence assay; ELISA, enzyme-linked immunosorbent assay, PCR, polymerase chain reaction.

## Methods

We performed a systematic literature search of the databases CINAHL, Cochrane EMBASE Elsevier, PubMed, Scopus and Web of Science on 16/Dec/2022, using the search term ("Anaplasma" OR "Anaplasmas" OR "HGE agent" OR "phagocytophil*" OR "E equi" OR "Ehrlichia equi" OR ("Ehrlichia" AND "granulocytotropic") OR "Anaplasmosis" OR "Anaplasmoses" OR (("Ehrlichiosis" OR "Ehrlichioses") AND "granulocyt*")) NOT ("Animals" NOT "Humans"), adapted to the search format of the different databases. A detailed description of the literature search is available in S2 Text. After removing duplicates by EndNote (Version X9.2, Clarivate Analytics) and manually, the publications were pre-screened by title and abstract, removing those not concerning HGA or not including the objectives of this study (data of clinical cases, including epidemiology, mode of transmission, diagnostic, treatment, outcome). A full-text review of the remaining publications was then performed excluding those not meeting the inclusion criteria, according to the systematic review protocol (concerning HGA and the objectives of the study, published in English, German, French, Italian or Spanish), as described in S1 Text. Publications that could neither be retrieved through the respective journals, nor by contacting libraries or the corresponding authors, were classified as 'not retrievable' and excluded. During the full-text review, the reference lists of the articles were screened for additional relevant publications not identified previously («snowball-search» strategy). We conducted a second literature search with the identical search strategy on 27/Apr/2023 to include the most recently published articles. From the finally identified eligible studies, the following data were extracted: author, title, year of publication, type of study, country of study, study period, number of HGA cases reported, age, sex, most likely country/province of acquisition/ infection, autochthonous or imported case, if imported: time between end of trip and

symptoms, country/province of diagnosis, risk factors for tick bite, year of acquisition/infection, pre-existing conditions, immunosuppression, pregnancy, symptomatic/asymptomatic infection, hospital admission, time between first symptoms and presentation to hospital/physician, duration of hospital stay, signs and symptoms, duration between fever and appropriate treatment, presumed vector of disease, history of tick bite/exposure, tick species, duration between bite and symptoms, diagnostic methods (serology, PCR, microscopy, culture, biopsy), grade of diagnostic certainty, *Anaplasma* species, coinfections, laboratory data, used drug(s) and treatment regimen(s), number of treated or untreated patients, complications, and outcome. Discrepancies and unclear cases were resolved by consulting a second reviewer. The probability of diagnostic certainty of the individual HGA cases was graded according to the diagnostic method used in the different studies, with PCR, culture and immunostaining of tissue having the highest (grade A+) and clinical diagnosis the lowest (grade D) evidence for a correct diagnosis (Table 3). The grading system was derived from the CDC case definition of

**Table 3. Grading system to judge the certainty of the correct diagnosis of HGA [17,20,21].**

| Diagnostic method | Description | Grade of diagnostic certainty | Case classification (provided illness clinically compatible with anaplasmosis) | Comment |
|---|---|---|---|---|
| PCR | Detection of *Anaplasma* spp. DNA in a clinical specimen via amplification of a specific target by polymerase chain reaction (PCR) assay | A+ | Direct evidence, confirmed diagnosis | High level of evidence, especially in the first week of illness and before start of antibiotics, mostly done from whole blood specimens, also possible in solid tissue and bone marrow specimens |
| Culture | Isolation of *Anaplasma* spp. from a clinical specimen in cell culture | A+ | Direct evidence, confirmed diagnosis | High level of evidence, especially in the first week of illness and before start of antibiotics, difficult to carry out, time demanding |
| Immunostaining of biopsy/autopsy tissue | Demonstration of anaplasmal antigen in a biopsy/autopsy sample by immunohistochemical methods | A+ | Direct evidence, confirmed diagnosis | High level of evidence, difficult to carry out, time demanding |
| Serology—IgG IFA, paired samples | Serological evidence of a fourfold rise in IgG-specific antibody titer to *A. phagocytophilum* antigens by indirect immunofluorescence assay (IFA) in paired serum samples (i.e. an acute phase sample [first week of infection] and a convalescent phase sample [2–4 weeks later]) | A | Indirect evidence, confirmed diagnosis | High level of evidence, serological gold standard, cross-reaction with other rickettsial diseases possible |
| Blood smear or buffy coat preparation microscopy | Identification of intracellular *morulae* in neutrophils or eosinophils by microscopic examination | B+ | Direct evidence, probable diagnosis | Intermediate level of evidence, easy and fast to carry out, examiner-dependent, likelihood of detection depends on level of *Anaplasma* in blood; limited specificity as morulae of *A. phagocytophilum* cannot be differentiated from morulae of *Ehrlichia ewingii* which also show a tropism for granulocytes |
| Serology—IgG IFA single sample or ELISA | Serological evidence of elevated IgG antibody reactive with *A. phagocytophilum* antigen by ELISA or IFA (CDC uses an IFA IgG cut-off of ≥1:64) | B | Indirect evidence, possible diagnosis | Low to intermediate level of evidence, no certain differentiation between acute and past infection possible, cross-reaction with other rickettsial diseases possible |
| Serology—IgM IFA or ELISA | Serological evidence of elevated IgM antibody reactive with *A. phagocytophilum* antigen by IFA, ELISA, or assays in other formats | C | Indirect evidence, possible diagnosis | Very low level of evidence, IgM detection lacks specificity to be useful as single means of diagnosis |
| Clinical diagnosis | Signs and symptoms compatible with anaplasmosis | D | Clinical diagnosis | The lowest level of evidence for correct diagnosis |

CDC, Centers for Disease Control and Prevention; DNA, deoxyribonucleic acid; ELISA, enzyme-linked immunosorbent assay; IFA, indirect fluorescent antibody assay; IgG, immunoglobulin G; IgM, immunoglobulin M; PCR, polymerase chain reaction.

HGA [17,20]. If a publication did not specify how individual cases were diagnosed (e.g. they were diagnosed by either high IgG or IgM IFA for *A. phagocytophilum*), we counted them in the category with the lower diagnostic evidence. To limit confounding, patients with coinfections and preexisting conditions were excluded from some analyses if it was not clear whether the symptoms, the laboratory abnormalities, complications, or the outcome were attributable to HGA.

For analysis, we divided the reviewed clinical cases into two categories: cases reported with individual data (CRID) and cases reported with non-individual data (CRNID), i.e. case series with cumulatively reported/pooled data and differentiated between HGA monoinfections and cases of HGA with coinfection(s). The used data extraction sheet is available in the S1 Table. We used the free online geographic application Mapchart (www.mapchart.net) to visualize the geographic distribution of HGA cases. Geographic data from studies included in the analysis were used to create the world and the US map (Figs 2 and 3). The world map was completed with seroprevalence data as summarized in the 2019 systematic review by Feng Wang et al. [22].

We included data from all investigated HGA cases in the analyses, regardless of the level of evidence for a correct diagnosis. In order to determine whether the analyses performed might give a distorted picture corresponding to the differences in diagnostic accuracy, we carried out a subgroup analysis, provided that a sufficiently large number of cases and thus data volume was available, and compared the analysis of the A+ and A cases with the analysis of all cases.

In addition to the performed descriptive analysis, summarizing data in percentages, medians, and ranges, a statistical analysis was performed to examine putative geographic differences in clinical manifestation of HGA using Student's t test (for normally distributed continuous

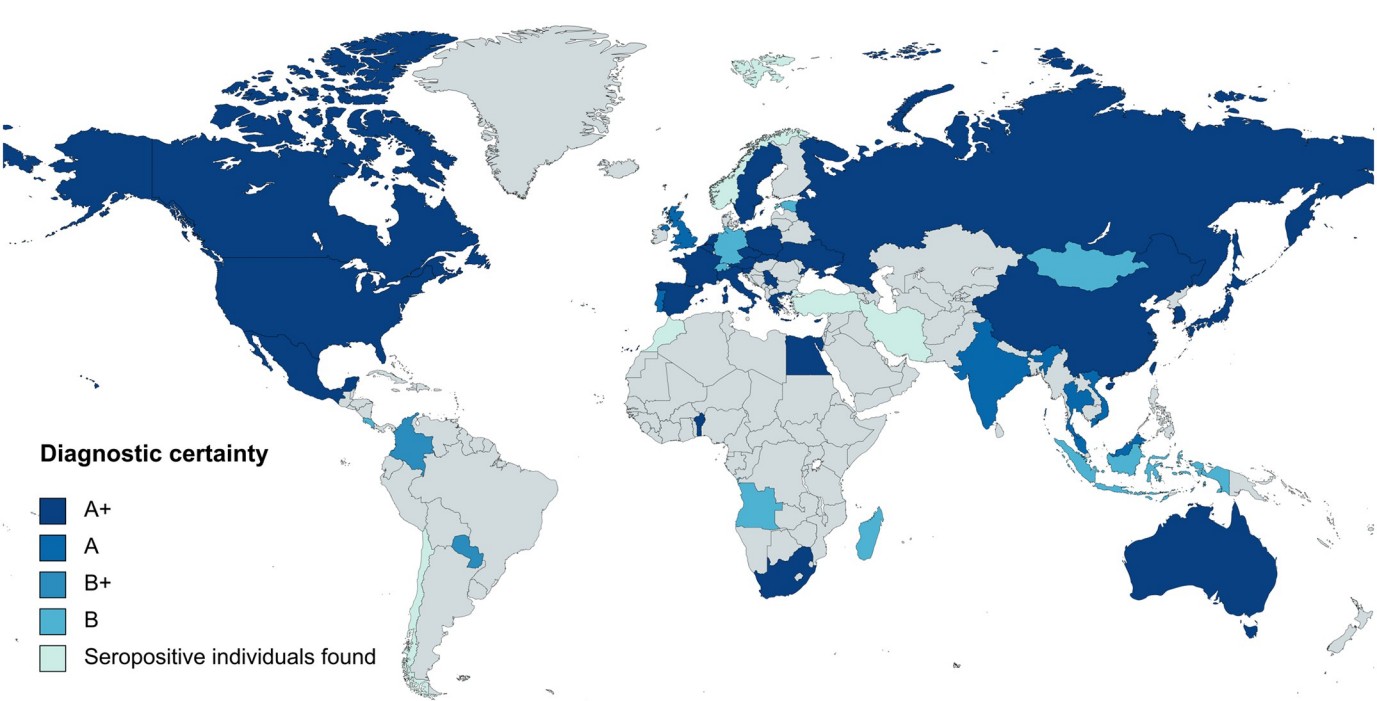

**Fig 2. Reported human granulocytotropic anaplasmosis cases by country.** A+, diagnosed by PCR, culture and/or immunostaining of biopsy/autopsy tissue; A, diagnosed by paired IgG IFA serology; B+, diagnosed by microscopy of blood smear or buffy coat preparation; B, diagnosed by single IgG IFA or ELISA serology. Created with mapchart.net.

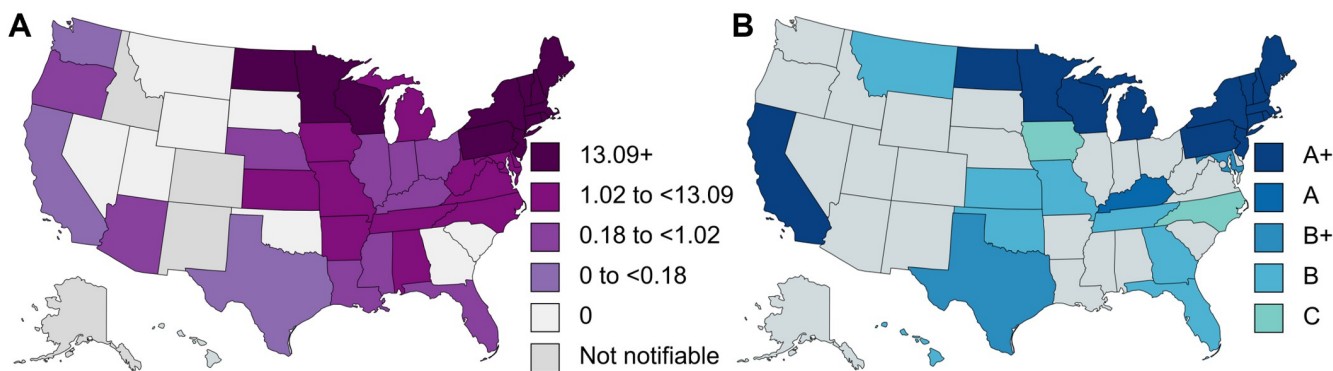

**Fig 3. Distribution of human granulocytotropic anaplasmosis in the United States of America.** (A) Annual incidence (per million population) of human granulocytotropic anaplasmosis according to national surveillance data in 2019 [3]; (B) Reviewed human granulocytotropic anaplasmosis cases according to diagnostic certainty: A+, diagnosed by PCR, culture and/or immunostaining of biopsy/autopsy tissue; A, diagnosed by paired IgG IFA serology; B+, diagnosed by microscopy of blood smear or buffy coat preparation; B, diagnosed by single-titer IgG IFA or ELISA serology; C, diagnosed by IgM serology. Created with mapchart.net.

data), Kruskal-Wallis test (for non-normally distributed continuous data), Pearson's chi-square test (for n≥5), or Fisher's exact test (for n<5) as appropriate. A p-value of <0.05 was considered to reflect statistical significance.

## Results

Our search identified 11'604 publications, of which 360 proved to be eligible for inclusion in the analysis (Fig 4). The reference lists of the included and excluded publications and the PRISMA checklist for systematic reviews are available in the S3 and S4 Text.

From the 360 publications included in the analysis, we extracted data on 3019 human anaplasmosis cases. The analysis of the data on signs and symptoms, laboratory finding, complications, treatment, and outcome was conducted separately for cases reported with individual data (CRID) and cases reported with non-individual data (CRNID). The analysis of the data on signs and symptoms, laboratory finding, complications, and outcome included in the manuscript is restricted to HGA monoinfections. The analysis of the data on epidemiology and antimicrobial treatment included mono- and coinfections. We also summarized the data on signs and symptoms, laboratory findings, complications, and outcome on CRID with coinfection(s) (S5 Text), but due to the data heterogeneity and the overall low numbers, we omitted the attempt of specific subgroup analyses. Although the analysis of the cases reported with non-individual data (CRNID; i.e. case series, cohorts) was limited, we nevertheless summarized the data analogously to CRID to allow a rough overview and comparison (S5 Text). Fig 5 shows the allocation of cases to the respective analysis groups.

### Epidemiology

Fig 6 and Table 4 show data on the number of publications reporting cases of human anaplasmosis, the number of CRID and CRNID, and the geographic origin of these reports from 1994 to 2023.

Table 5 lists the number of CRID and CRNID reported from the different regions of the world. Fig 2 shows the worldwide distribution of reported HGA cases. Fig 3 shows the

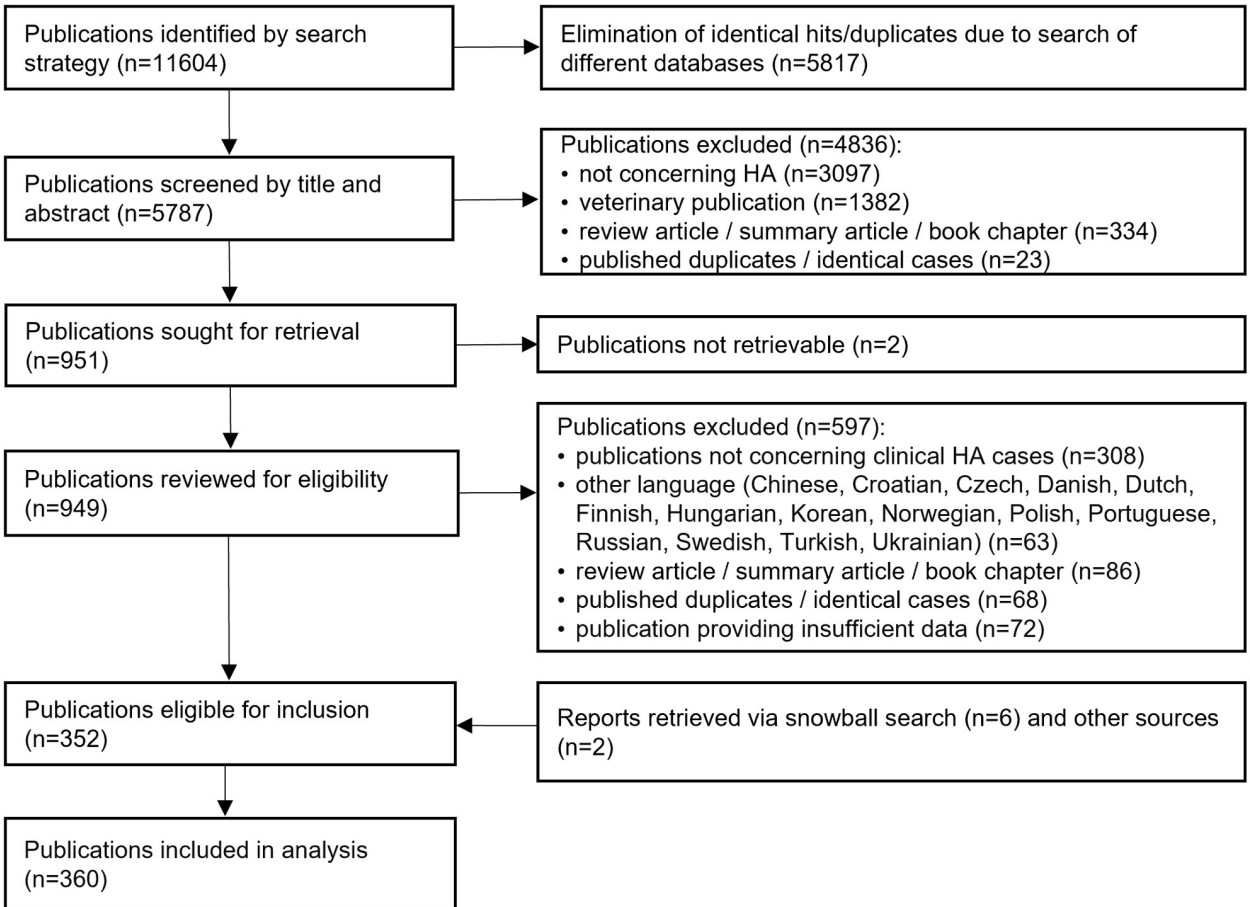

**Fig 4. Flow diagram of search and selection of eligible publications.** HA, human anaplasmosis.

distribution of HGA cases in the USA according to national surveillance data and according to the data analyzed for this review.

Of the 360 analyzed publications, 10 publications reported a total number of 68 non-*phagocytophilum* human anaplasmosis cases (Table 6). Reported concurrently present co-infections of HGA patients with other pathogens are listed in Table 7. Data on the diagnostic methods used to diagnose HGA was available for 609 CRID and 2068 CRNID (Table 8).

In 160 cases, the cells in which morulae were found by microscopy were specified. In 151 (94.4%) cases, morulae were described in granulocytes, in 4 (2.5%) cases (PCR confirmed) morulae were described in monocytes [26–29], in 3 (1.9%) cases (1 PCR confirmed) the morulae were described in leukocytes not specified, in 1 (0.6%) case (PCR confirmed) morulae were described in not specified leukocytes, erythrocytes and platelets [30], and in 1 (0.6%) case morulae were described in stromal macrophages (not-PCR confirmed) [31].

Table 9 shows the analysis of PCR-confirmed HGA cases in which also data on microscopic results was available regarding the reported presence of *morulae*.

The cell-line used for culture was specified for 19 cases: culture was performed in HL-60 cells in all 19 cases and in one case culture was additionally performed in leukemic cells isolated from the patient [32]. For one case suffering from cranial nerve neuritis, positive culture was not performed from blood, but from CSF [33].

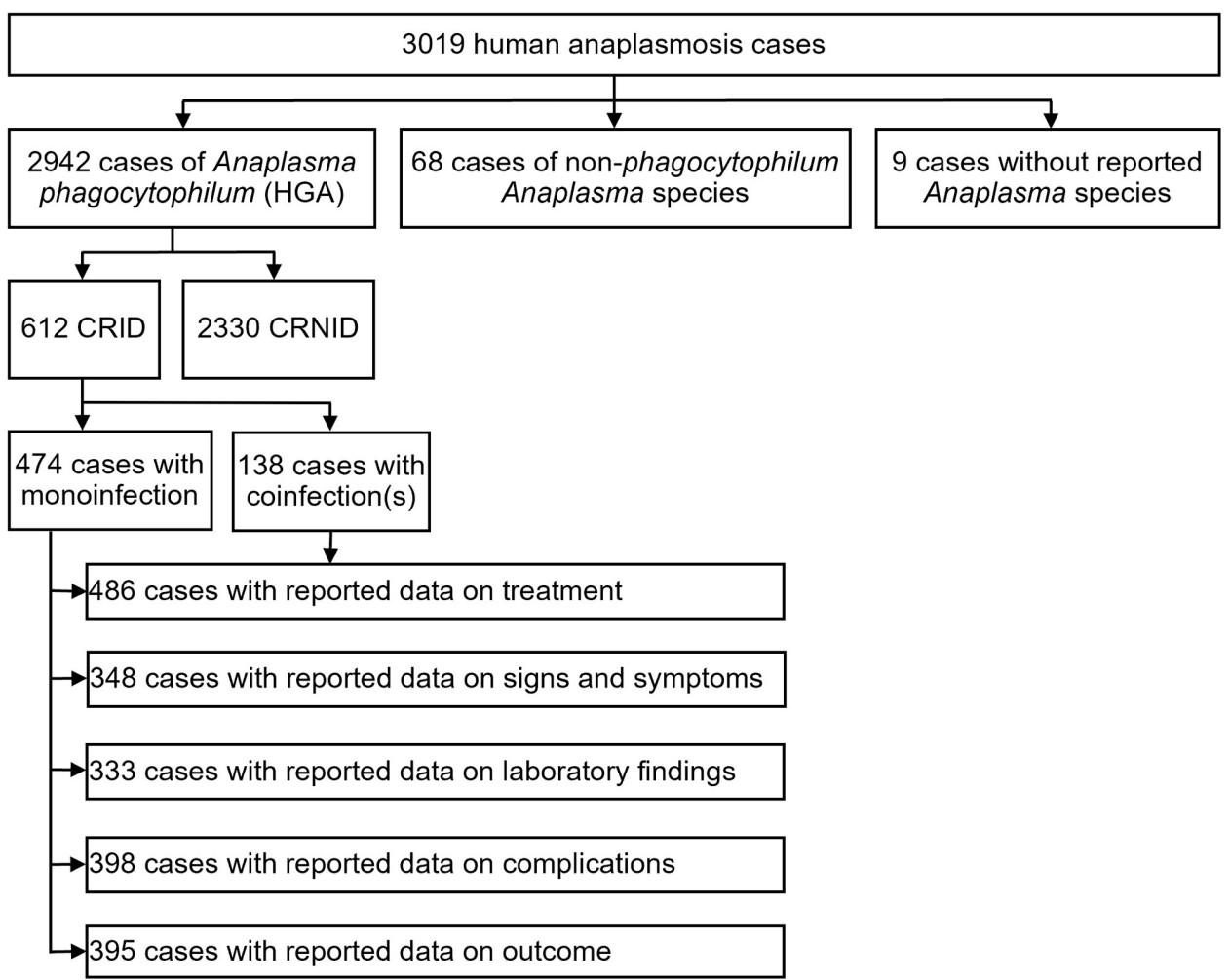

**Fig 5. Allocation of the reviewed human anaplasmosis cases to the respective analysis groups.** CRID, cases reported with individual data; CRNID, cases reported with non-individual data; HGA, human granulocytotropic anaplasmosis.

### Analysis of HGA infection cases reported with individual data (CRID)

For 563 cases, the patient's sex was reported: 332 (59.0%) were male, 231 (41.0%) were female. For 528 cases (319 male, 209 female), sex and age were reported: the median age (range) of male and female patients was 57 (<1–95), and 56 years (<1–88), respectively. 45 cases (7.4%) were reported to be immunocompromised/ immunosuppressed. 21 were on immunosuppressive medication, six had hematological malignancies, eight cases were asplenic, two hyposplenic (one of them additionally being on immunosuppressive medication). Two cases had a myelodysplastic syndrome, with one of them being splenectomized. Four cases had end-stage renal disease, with one of them additionally suffering from end-stage liver disease. Two cases were HIV positive with a below normal $CD_4$ count.

Table 10 shows the analysis of HGA patients' characteristics according to geographic region.

The suspected route of transmission was tick-borne in 584 (95.4%) cases. Among 348 patients actively assessed for the history of a tick bite, 250 (71.8%) recalled a tick bite. In 30

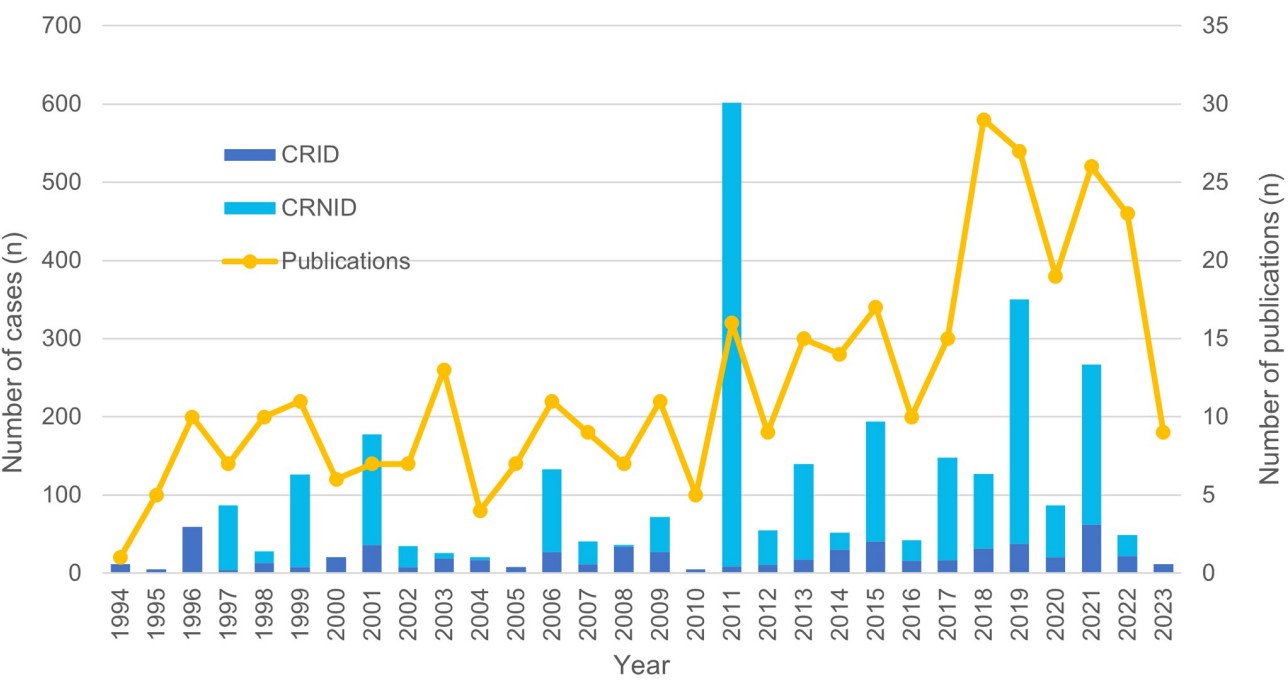

**Fig 6. Number of publications reporting cases of human granulocytotropic anaplasmosis and the respective number of cases published 1994–2023.** CRID, cases reported with individual data; CRNID, cases reported with non-individual data.

patients, the culprit tick species was reported: 26 *Ixodes* spp. (18 *I. scapularis*, 6 *I. ricinus*, 1 *I. nipponensis*, 1 unspecified *Ixodes* sp.) and four *Dermacentor* spp. In 225 (94.5%) of 238 cases with respectively available data, outdoor activities were reported as risk factor for HGA. For 87 cases the outdoor activities were specified: 30 (34.5%) were occupational and 57 (65.5%) were recreational. Other reported routes of HGA transmission included: transfusion of blood products in 11 cases (Table 11), a case cluster of 9 secondary cases resulting from contact to body fluids of a single index patient [34], 5 single cases resulting from contact to animal blood [35–37], and vertical transmission (most likely transplacental transmission) in 1 case [38]. Two cases were suspected to be infected by working in a veterinary clinic, but the exact route of infection was unclear [39].

17 cases each of HGA were reported during pregnancy (Table 12) and in association with international travel (Table 13).

Of the 474 HGA monoinfection CRID, data on whether they were symptomatic or asymptomatic was available for 469: 451 (96.2%) were symptomatic, 18 (3.8%) were asymptomatic. Data on the incubation period in case of tick-borne transmission was available for 88 cases: median 7 days (range 1–30 days). Data on the incubation period in case of transmission by blood product transfusion was available for 9 patients: median 9 days (range 4–11 days). Hospitalization was reported for 267 (59.2%) of the symptomatic HGA cases. Data on the length of hospitalization was available for 54 patients: median 8 days (range 2–37 days).

## Signs and symptoms

For 348 of the 474 HGA monoinfection CRID (Fig 5), data on signs and symptoms was available. Fig 7 shows the frequency of the most commonly reported symptoms.

**Table 4. Number of publications reporting human anaplasmosis cases and number of reported cases by country 1994–2023.**

| Country | Number of publications reporting human anaplasmosis cases n (%) | Number of reported human anaplasmosis cases n (%) |
|---|---|---|
| USA | 212 (58.9) | 1085 (35.9) |
| China | 14 (3.9) | 478 (15.8) |
| Poland | 14 (3.9) | 289 (9.6) |
| South Korea | 14 (3.9) | 62 (2.1) |
| Slovenia | 12 (3.3) | 135 (4.5) |
| Canada | 11 (3.1) | 37 (1.2) |
| Austria | 9 (2.5) | 18 (<1) |
| Belgium | 6 (1.7) | 444 (14.7) |
| France | 6 (1.7) | 44 (1.5) |
| Italy | 6 (1.7) | 13 (<1) |
| Spain | 5 (1.4) | 13 (<1) |
| Switzerland | 5 (1.4) | 23 (<1) |
| Slovakia | 4 (1.1) | 32 (1.1) |
| Czech Republic | 3 (<1) | 24 (<1) |
| Germany | 3 (<1) | 7 (<1) |
| Greece | 3 (<1) | 8 (<1) |
| Sweden | 3 (<1) | 11 (<1) |
| Cyprus | 2 (<1) | 2 (<1) |
| Israel | 2 (<1) | 2 (<1) |
| Japan | 2 (<1) | 6 (<1) |
| Mexico | 2 (<1) | 87 (2.9) |
| Netherlands | 2 (<1) | 6 (<1) |
| Russia | 2 (<1) | 107 (3.5) |
| Taiwan | 2 (<1) | 12 (<1) |
| Venezuela | 2 (<1) | 32 (1.1) |
| Australia | 1 (<1) | 8 (<1) |
| Colombia | 1 (<1) | 1 (<1) |
| Egypt | 1 (<1) | 6 (<1) |
| Estonia | 1 (<1) | 1 (<1) |
| French Guiana | 1 (<1) | 1 (<1) |
| Iran | 1 (<1) | 10 (<1) |
| Mongolia | 1 (<1) | 1 (<1) |
| Norway | 1 (<1) | 1 (<1) |
| Paraguay | 1 (<1) | 4 (<1) |
| Portugal | 1 (<1) | 2 (<1) |
| Serbia | 1 (<1) | 3 (<1) |
| Türkiye | 1 (<1) | 1 (<1) |
| United Kingdom | 1 (<1) | 1 (<1) |
| Ukraine | 1 (<1) | 2 (<1) |
| Total | 360 (100) | 3019 (100) |

USA, United States of America.

**Table 5. Number of human granulocytotropic anaplasmosis cases by geographic region.**

| Geographic region | Number of CRID n (%) | Number of CRNID n (%) |
|---|---|---|
| North America | 312 (51.0) | 898 (38.5) |
| Europe | 210 (34.3) | 860 (36.9) |
| Asia | 83 (13.6) | 558 (23.9) |
| Africa | 3 (<1) | 6 (<1) |
| Australia | - | 8 (<1) |
| South America | 4 (<1) | - |
| Total | 612 (100) | 2330 (100) |

CRID, cases reported with individual data; CRNID, cases reported with non-individual data.

**Table 6. Reported cases of human anaplasmosis due to non-*phagocytophilum Anaplasma* species (n = 68).**

| *Anaplasma* species | Number of cases (n) | Country of infection | Year of publication | Mode of diagnosis (n patients) | Clinical features (n patients) | Comment | Ref. |
|---|---|---|---|---|---|---|---|
| *Anaplasma platys* | 30 | Venezuela | 1997 | Microscopy | N.r. | | [23] |
| | 1 | Grenada | 2013 | PCR | Malaise, headache, arthralgia, generalized muscle fasciculations, tonic-clonic seizures / status epilepticus, left-sided weakness, fainting, vertigo, tremors, ataxia, memory loss | Veterinarian with coinfections: *Bartonella henselae, Candidatus* Mycoplasma haematoparvum | [8] |
| | 2 | Venezuela | 2014 | PCR | Headache (2/2), myalgia (2/2), arthralgia (1/2), anorexia (1/2), weakness (1/2), anemia (1/2), morulae in thrombocytes (2/2) | | [9] |
| | 1 | Paraguay | 2016 | Microscopy | Fever, petechial rash, myalgia, arthralgia | | [24] |
| *Anaplasma bovis* | 1 | China | 2019 | PCR | Fever, rash, chills, headache, myalgia, lymphadenopathy | | [10] |
| | 2 | China | 2022 | PCR (1/2), serology (1/2) | Fever (2/2), weakness (2/2), anorexia (2/2), thrombocytopenia (2/2), myalgia + headache (1/2), headache (1/2), rash (1/2), diarrhea (1/2), elevated liver enzymes (1/2) | | [12] |
| *Anaplasma capra* | 28 | China | 2015 | PCR (28/28), culture (3/28), serology (22/28) | All symptomatic, specific data only for 5 patients: fever (5/5), headache (5/5), lymphadenopathy (5/5), meningoencephalitic symptoms (1/5) | | [11] |
| *Anaplasma ovis* | 1 | Iran | 2014 | PCR | Asymptomatic | In the same publication description of 9 other asymptomatic cases, diagnosed by PCR as *Anaplasma* spp., but not specified by sequencing | [25] |
| | 1 | Cyprus | 2010 | PCR | Fever, lymphadenopathy, hepatosplenomegaly, thrombocytopenia, mild anemia, elevated liver enzymes | | [7] |
| *Candidatus* Anaplasma sparouinense | 1 | French Guiana | 2022 | PCR | Anemia, fever, myalgia, headache, epistaxis; inclusion bodies in erythrocytes | Splenectomized patient, possible *Coxiella burnetii* coinfection | [13] |

N.r. none/not reported; PCR, polymerase chain reaction; Ref., reference.

**Table 7. Reported coinfections in human granulocytotropic anaplasmosis.**

| Reported coinfection pathogen(s)* | Number of coinfections among HGA CRID [n = 612] n (%) | Number of coinfections among HGA CRNID [n = 2330] n (%) | Number of coinfections among all HGA cases [n = 2942] n (%) |
|---|---|---|---|
| **Tick-borne pathogens** | | | |
| *Borrelia burgdorferi* sensu lato | 72 (11.8) | 247 (10.6) | 319 (10.8) |
| Tick-borne encephalitis virus (TBEV) [#] | 5 (<1) | 64 (2.7) | 69 (2.3) |
| *Borrelia burgdorferi* s.l. + Tick-borne encephalitis virus (TBEV) [#] | 1 (<1) | 14 (<1) | 15 (<1) |
| ***Borrelia burgdorferi* s.l. + *Bartonella henselae*[†]** | - | 8 (<1) | 8 (<1) |
| ***Borrelia burgdorferi* s.l. + *Babesia* spp.** | 3 (<1) | 3 (<1) | 6 (<1) |
| ***Borrelia burgdorferi* s.l. + Powassan virus** | 1 (<1) | - | 1 (<1) |
| ***Borrelia burgdorferi* s.l. + *Rickettsia* spp.** | 1 (<1) | - | 1 (<1) |
| *Babesia* spp. | 10 (1.6) | 15 (<1) | 25 (<1) |
| Severe fever with thrombocytopenia syndrome virus (SFTSV) | 10 (1.6) | 1 (<1) | 11 (<1) |
| *Rickettsia* spp. | 3 (<1) | 1 (<1) | 4 (<1) |
| *Ehrlichia chaffeensis* | 3 (<1) | - | 3 (<1) |
| *Coxiella burnetti* | 1 (<1) | - | 1 (<1) |
| Novel Haseki tick virus (HSTV) | 1 (<1) | - | 1 (<1) |
| Unspecified bunyavirus infection | - | 1 (<1) | 1 (<1) |
| **Other arthropod-borne pathogens** | | | |
| *Orienta tsutsugamushi* | 15 (2.4) | 3 (<1) | 18 (<1) |
| **Dengue virus + enteroaggregative *Escherichia coli*** | 1 (<1) | - | 1 (<1) |
| Chikungunya virus | 1 (<1) | - | 1 (<1) |
| **Non-vector-borne pathogens** | | | |
| *Toxoplasma gondii* | 3 (<1) | - | 3 (<1) |
| Epstein-Bar virus | 2 (<1) | - | 2 (<1) |
| Influenza B virus | 1 (<1) | - | 1 (<1) |
| Unspecified coinfection | 4 (<1) | - | 4 (<1) |
| Total | 138 (22.5) | 357 (15.3) | 495 (16.8) |

HGA, human granulocytotropic anaplasmosis; CRID, cases reported with individual data; CRNID, cases reported with non-individual data.

* Note that the list of coinfections is based on the coinfections reported by the authors in the original publications, which, however, often cannot be checked for validity due to a lack of information on the corresponding diagnostics and therefore do not allow a conclusive assessment.

[#] TBEV only occurs in Europe and Asia. Therefore, calculating the co-infection rates for HGA + TBEV with the corresponding region-specific denominator of the cases reported from Europe and Asia results in co-infection rates of 1.7% for CRID, 10.8% CRNID, and 7.8% CRID + CRNID. The co-infection rates of HGA, + *B. burgdorferi* s.l. + TBEV would be <1% for CRID, 2.4% for CRNID, and 1.7% for CRID + CRIND.

[†] Note that *Bartonella henselae* is not generally considered a tick-borne pathogen (see Discussion).

The presence of a rash was reported in 47 (13.5%) cases. The reported rash morphology was diverse with maculopapular, erythematous, and petechial being most common. In a few cases, rashes were reported to be itchy or painful.

Table 14 shows the analysis of HGA patients' symptoms and signs according to geographic region.

## Laboratory findings

Of the 474 HGA monoinfection CRID, data on laboratory findings was available for 333 cases (Fig 5). Fig 8 shows the frequency and Table 15 the median values and ranges of the reported abnormal laboratory findings.

**Table 8. Diagnostic methods used to diagnose human granulocytotropic anaplasmosis.**

| Diagnostic method | Level of diagnostic certainty* | HGA CRID [n = 609] | | HGA CRIND [n = 2068] | |
|---|---|---|---|---|---|
| | | Number of cases tested positive by this method n (%)# | Number of cases for which this level of diagnostic certainty was the highest n (%) | Number of cases tested positive by this method n (%)# | Number of cases for which this level of diagnostic certainty was the highest n (%) |
| PCR | A+ | 335 (55.0) | 337 (55.3) | 862 (41.7) | 871 (42.1) |
| Culture | | 24 (3.9) | | 55 (2.7) | |
| Immunostaining of biopsy tissue | | 4 (<1) | | - | |
| Serology–IgG IFA, paired samples | A | 121 (19.9) | 40 (6.6) | 327 (15.8) | 251 (12.1) |
| Microscopy | B+ | 187 (30.7) | 68 (11.2) | 135 (6.5) | 34 (1.7) |
| Serology–IgG IFA, single sample | B | 259 (42.5) | 141 (23.2) | 570 (27.6) | 529 (25.6) |
| Serology–IgG ELISA, single sample | | 3 (<1) | | 106 (5.1) | |
| Serology–IgM IFA or ELISA | C | 22 (3.6) | 20 (3.3) | 44 (2.1) | 357 (17.2) |
| Serology–method not specified | | 15 (2.5) | | 393 (19.0) | |
| Clinical diagnosis | D | 3 (<1) | 3 (<1) | 26 (1.3) | 26 (1.3) |

HGA, human granulocytotropic anaplasmosis; CRID, cases reported with individual data; CRNID, cases reported with non-individual data; ELISA, enzyme-linked immunosorbent assay; IFA, indirect fluorescent antibody assay; PCR, polymerase chain reaction.

* A+, diagnosed by PCR, culture and/or immunostaining of biopsy/autopsy tissue; A, diagnosed by paired IgG IFA serology; B+, diagnosed by microscopy of blood smear or buffy coat preparation [of note: *A. phagocytophilum* cannot be differentiated from *Ehrlichia ewingii*, by microscopy as morulae of both species show a tropism for granulocytes]; B, diagnosed by single IgG IFA or ELISA serology; C, diagnosed by IgM serology; D, clinically diagnosed.

# in many cases a combination of diagnostic tests was used to establish the diagnosis. Thus, the number of positive test results exceeds the number of cases.

Table 16 shows the analysis of abnormal laboratory findings in HGA patients according to geographic region.

## Complications

Complications were reported in 161 (40.5%) of 398 HGA monoinfection CRID (Fig 9). Note: exacerbation of an underlying medical condition was not considered a complication of HGA. The analysis of the occurrence of complications in HGA according to geographic region can be found in Table 17.

**Table 9. Presence of *morulae* in PCR-confirmed human granulocytotropic anaplasmosis according to geographic region (n = 194).**

| Result of microscopy in PCR-confirmed cases of HGA [n = 194] | Geographic region | | | p-value | | |
|---|---|---|---|---|---|---|
| | North America [n = 119] n (%) | Europe [n = 33] n (%) | Asia [n = 42] n (%) | North America vs. Europe | North America vs. Asia | Europe vs. Asia |
| *Morulae* present | 92 (77) | 13 (39) | 13 (31) | <0.0001 | <0.0001 | 0.6044 |
| *Morulae* not present | 27 (23) | 20 (61) | 29 (69) | | | |

HGA, human granulocytotropic anaplasmosis; PCR, polymerase chain reaction; vs., versus.

Note: in the absence of corresponding data from Africa and South America, respective analyses were not possible.

**Table 10. Patients' characteristics in human granulocytotropic anaplasmosis (with A+ and A diagnostic certainty) according to geographic region.**

| Patients' characteristics | | Geographic region | | | | | p-value | | |
|---|---|---|---|---|---|---|---|---|---|
| | | North America [n = 186] % or median [IQR] (n) | Europe [n = 66] % or median [IQR] (n) | Asia [n = 44] % or median [IQR] (n) | Africa [n = 1] % or median [IQR] (n) | South America [n = 0] % or median [IQR] (n) | North America vs. Europe | North America vs. Asia | Europe vs. Asia |
| **Age median (IQR)** | | 64 [48–75] (179) | 46 [34–58] (51) | 70 [53–79] (41) | 57 [N.c.] (1) | - | | | |
| **Age group** | ≤20 | 4 (8) | 14 (7) | 2 (1) | - | - | | | |
| | 21–40 | 10 (18) | 20 (10) | 10 (4) | - | - | | | |
| | 41–60 | 24 (42) | 46 (24) | 27 (11) | 100 (1) | - | | | |
| | 60–80 | 48 (85) | 20 (10) | 39 (16) | - | - | | | |
| | >80 | 14 (24) | - | 22 (9) | - | - | | | |
| | N.r. | 9 | 15 | 3 | - | - | | | |
| **Sex** | Male | 61 (108) | 61 (34) | 41 (17) | 100 (1) | - | | | |
| | Female | 39 (70) | 39 (22) | 59 (24) | - | - | | | |
| | N.r. | 8 | 10 | 3 | - | - | | | |
| **Immunodeficiency/-suppression*** | Present | 13 (24) | 5 (3) | - | - | - | 0.066 | **0.006** | 0.273 |
| | Absent | 87 (162) | 95 (63) | 100 (44) | 100 (1) | - | | | |
| **Hospitalization** | Yes | 69 (123) | 76 (39) | 77 (33) | - | - | 0.611 | 0.396 | 0.905 |
| | No | 31 (56) | 24 (14) | 23 (10) | 100 (1) | - | | | |
| | N.r. | 7 | 13 | 1 | 0 | - | | | |

IQR, interquartile range; N.r., not reported; N.c., not calculated.; vs., versus.

* Definition of immunodeficiency/-suppression: (i) any medical treatment aiming at and leading to impaired immune defence (i.e. corticosteroids above substitution dosage, azathioprine, methotrexate, chemotherapy, biologics a.o.) and (ii) any clinical condition likely to impair immune defence (i.e. hematological malignancies and disorders, asplenia, HIV-infection with a below normal CD4-T-cell count, uncontrolled diabetes mellitus, end-stage kidney or liver disease).

## Treatment

Data on antimicrobial treatment was available for 486 of the analyzed 612 HGA CRID. 391 cases (80.5%) received appropriate* antimicrobial treatment (*see Table 18), including 149 cases (38.1%) in which the initial empirical treatment was appropriate* (*see Table 18), 123 cases (31.5%), in which the initial empirical treatment was inappropriate* (*see footer of Table 18) but which were later switched to an appropriate treatment regimen, and 119 cases (30.4%) in which it was unclear whether the initial empirical treatment was appropriate or whether the treatment was later switched. 95 cases (19.5%) did not (never) receive appropriate antimicrobial treatment.

Data on the time window between first administration of appropriate antimicrobial treatment and the resolution of fever was available for 86 cases: median time 1 day (range <1–9 days). In four cases, treatment was only started after the fever had already subsided. In the majority of cases doxycycline was given in the standard dose of 200 mg per day or weight adapted for children. The overall median duration of antimicrobial treatment was 14 days (range 1–42 days).

Data on antimicrobial treatment in different geographic regions is shown in Table 19. Data on the time windows between onset of symptoms, presentation to a medical facility and administration of appropriate antimicrobial treatment is shown in Table 20.

**Table 11. Reported cases of blood product transfusion-transmitted human granulocytotropic anaplasmosis (n = 11).**

| No. | Year of publication | Age of patient (years) | Sex of patient | Country of infection | Pre-existing medical conditions | Immuno-suppressive treatment | Culprit blood product | Diagnosis of HGA established by | Antimicrobial therapy | Time between onset of fever and appropriate treatment (days) | Complications | Outcome | Ref. |
|---|---|---|---|---|---|---|---|---|---|---|---|---|---|
| 1 | 1999 | 75 | Male | USA | Rheumatoid arthritis | N.r. | RBC | PCR | Doxycycline | N.r. | N.r. | Survived | [40] |
| 2 | 2008 | 68 | Male | USA | Psoriatic arthritis, ankylosing spondylitis | Corticosteroids | RBC | PCR | Doxycycline, cefazolin, piperacillin tazobactam, levofloxacin, trimethoprim sulfamethoxazole | N.r. | N.r. | Survived | [41] |
| 3 | 2011 | 36 | Female | Slovenia | N.r. | N.r. | RBC | PCR | N.r. | N.r. | ARDS | N.r. | [42] |
| 4 | 2012 | 81 | Female | USA | Rheumatoid arthritis | Prednisone, methotrexate | Leukoreduced RBC | PCR | N.r. | N.r. | Multi-organ failure, DIC | Survived | [43] |
| 5 | 2012 | 51 | Female | USA | Multiple myeloma | Stem cell transplantation | Unspecified blood transfusion | PCR | Doxycycline, imipenem, vancomycin | 7 | Nephrotic syndrome | Survived | [43] |
| 6 | 2012 | 36 | Female | Slovenia | Pregnancy complicated by preeclampsia | N.r. | RBC | PCR | Doxycycline, amoxicillin clavulanate, gentamicin, metronidazole, imipenem, azithromycin, vancomycin, piperacillin tazobactam, daptomycin | 6 | ARDS | Survived | [44] |
| 7 | 2013 | 64 | Male | USA | COPD exacerbation | High-dose corticosteroids | Leukoreduced RBC | PCR | Doxycycline, azithromycin, vancomycin, piperacillin tazobactam | 8 | Hypercarbic respiratory failure, non-invasive ventilation | Survived | [45] |
| 8 | 2014 | 41 | Male | USA | N.r. | N.r. | Leukoreduced platelets | PCR | N.r. | N.r. | N.r. | Died of trauma unrelated to HGA | [46] |
| 9 | 2015 | 34 | Female | USA | β-thalassemia, 26th week of pregnancy | N.r. | Leukoreduced RBC | PCR | Rifampicin | 25 | N.r. | Survived | [47] |
| 10 | 2016 | 78 | Female | USA | Coronary artery disease, type 2 diabetes | N.r. | Leukoreduced platelets | PCR | Doxycycline, vancomycin, piperacillin tazobactam | 1 | Cardiopulmonary decompensation, acute kidney injury | Survived | [48] |

(*Continued*)

**Table 11.** (Continued)

| No. | Year of publication | Age of patient (years) | Sex of patient | Country of infection | Pre-existing medical conditions | Immuno-suppressive treatment | Culprit blood product | Diagnosis of HGA established by | Antimicrobial therapy | Time between onset of fever and appropriate treatment (days) | Complications | Outcome | Ref. |
|---|---|---|---|---|---|---|---|---|---|---|---|---|---|
| 11 | 2018 | 78 | Male | USA | Coronary artery disease, heart failure, type 2 diabetes, chronic kidney disease | N.r. | Leukoreduced RBC | PCR | Doxycycline, vancomycin, piperacillin tazobactam | 4 | Multi-organ failure, respiratory failure, refractory shock | Died | [16] |

ARDS, acute respiratory distress syndrome; COPD, chronic obstructive pulmonary disease; DIC, disseminated intravascular coagulopathy; HGA, human granulocytotropic anaplasmosis; N.r., none/not reported; PCR, polymerase chain reaction; RBC, red blood cells; Ref, reference; USA, United States of America.

**Table 12. Reported cases of human granulocytotropic anaplasmosis in pregnancy (n = 17).**

| No. | Year of publication | Age of patient (years) | Country of infection | Pre-existing medical conditions | Week of pregnancy or trimester | Level of diagnostic certainty# | Coinfections | Antimicrobial therapy | Duration between onset of fever and appropriate treatment (days) | Complications | Outcome of the mother | Outcome of the child | Ref. |
|---|---|---|---|---|---|---|---|---|---|---|---|---|---|
| 1 | 1998 | 30 | USA | N.r. | 25 | A+ | *B. burgdorferi* | Rifampicin, cefuroxime | N.r. | N.r. | Survived | No evidence of infection | [49] |
| 2 | 1998 | 23 | USA | N.r. | 36 | A+ | N.r. | Rifampicin | N.r. | N.r. | Survived | No evidence of infection | [49] |
| 3 | 1998 | 35 | USA | N.r. | 39 | A+ | N.r. | Doxycycline, clindamycin, gentamicin | 6 | N.r. | Survived | Perinatal transmission, symptomatic, cure after treatment with doxycycline | [38] |
| 4 | 2002 | 38 | USA | N.r. | 34 | B | N.r. | Doxycycline, amoxicillin, gentamicin, erythromycin | N.r. | N.r. | Survived | No evidence of infection | [50] |
| 5 | 2007 | 36 | USA | N.r. | 10 | A+ | N.r. | Rifampicin | N.r. | N.r. | Survived | No evidence of infection | [51] |
| 6 | 2007 | 28 | USA | N.r. | 34 | A+ | N.r. | Rifampicin | N.r. | N.r. | Survived | No evidence of infection | [51] |
| 7 | 2007 | 29 | USA | N.r. | 34 | A | N.r. | Doxycycline | N.r. | N.r. | Survived | No evidence of infection | [51] |
| 8 | 2007 | 25 | USA | N.r. | 26 | A+ | N.r. | Rifampicin | N.r. | N.r. | Survived | No evidence of infection | [51] |
| 9 | 2007 | 30 | USA | N.r. | 24 | A | N.r. | Unspecified beta-lactam antibiotic | N.r. | N.r. | Survived | No evidence of infection | [51] |
| 10 | 2008 | 22 | USA | N.r. | 7 | D | N.r. | Doxycycline | 3 | Miscarriage, consumptive coagulopathy | Survived | Miscarriage | [52] |
| 11 | 2009 | 36 | Czech Republic | N.r. | First trimester | A+ | *B. garinii* | Penicillin | N.r. | Miscarriage | Survived | Miscarriage | [53] |
| 12 | 2009 | 33 | Czech Republic | N.r. | First trimester | A+ | *B. garinii* | Penicillin | N.r. | Miscarriage | Survived | Miscarriage, *B. garinii* was isolated from placenta | [53] |
| 13 | 2009 | 37 | Czech Republic | N.r. | First trimester | A+ | *B. garinii* | Penicillin | N.r. | N.r. | Survived | Healthy twins, one of whom had positive antibodies against *B. burgdorferi* | [53] |
| 14 | 2011 | 28 | USA | N.r. | 12 | B | N.r. | Rifampicin | N.r. | N.r. | Survived | No evidence of infection | [54] |

(*Continued*)

**Table 12.** (Continued)

| No. | Year of publication | Age of patient (years) | Country of infection | Pre-existing medical conditions | Week of pregnancy or trimester | Level of diagnostic certainty# | Coinfections | Antimicrobial therapy | Duration between onset of fever and appropriate treatment (days) | Complications | Outcome of the mother | Outcome of the child | Ref. |
|---|---|---|---|---|---|---|---|---|---|---|---|---|---|
| 15 | 2015 | 34 | USA | β-thalassemia with anemia requiring recent RBC transfusion | 32 | A+ | N.r. | Rifampicin | 25 | N.r. | Survived | No evidence of infection | [47] |
| 16 | 2015 | 34 | USA | Crohn's disease (on azathioprine) | 18 | B+ | N.r. | Rifampicin, amoxicillin clavulanate, ertapenem | N.r. | Hemodynamic instability, pneumonitis | Survived | No evidence of infection | [55] |
| 17 | 2021 | 31 | USA | N.r. | 24 | A+ | *B. burgdorferi* | Amoxicillin | N.r. | Impaired consciousness | Survived | No evidence of infection | [56] |

*B. burgdorferi*, *Borrelia burgdorferi*; *B. garinii*, *Borrelia garinii*; N.r., none/not reported; RBC, red blood cell; Ref., reference; USA, United States of America.

# A+, diagnosed by PCR, culture and/or immunostaining of biopsy/autopsy tissue; A, diagnosed by paired IgG IFA serology; B+, diagnosed by microscopy of blood smear or buffy coat preparation; B, diagnosed by single IgG IFA or ELISA serology; D, clinical diagnosis.

**Table 13. Reported cases of travel-related human granulocytotropic anaplasmosis (n = 17).**

| No. | Year of publication | Number of cases (n) | Continent of infection | Visited countries | Country of diagnosis | Level of diagnostic certainty of HGA* | Coinfections | Ref. |
|---|---|---|---|---|---|---|---|---|
| 1 | 1996 | 1 | Europe | Slovenia | Switzerland | B | *B. burgdorferi s.l.* | [57,58] |
| 2 | 2006 | 1 | Europe | Austria, Slovenia | Germany | C | *B. burgdorferi s.l.* | [59] |
| 3 | 2007 | 1 | Europe | Czech Republic | Spain | B+ | N.r. | [60] |
| 4 | 2009 | 1 | Europe | Netherlands | USA | B+ | N.r. | [61] |
| 5 | 2014 | 1 | Europe | Scotland | Germany | A | N.r. | [62] |
| 6 | 2016 | 1 | Europe | France | Belgium | C | N.r. | [63] |
| 7 | 2016 | 1 | North America | USA | Austria | A+ | N.r. | [64] |
| 8 | 2017 | 1 | North America | USA | Israel | A+ | N.r. | [65] |
| 9 | 2021 | 1 | Africa | Angola | Spain | B | N.r. | [66] |
| 10 | 2021 | 1 | Asia | Vietnam | Spain | A | Dengue virus, EAEC | [66] |
| 11 | 2021 | 1 | Asia | Indonesia | Spain | B | N.r. | [66] |
| 12 | 2021 | 1 | Asia | Malaysia, Singapore, Thailand | Spain | A+ | Chikungunya virus | [66] |
| 13 | 2021 | 1 | Africa | Madagascar | Spain | B | N.r. | [66] |
| 14 | 2021 | 1 | South America | Costa Rica | Spain | B | N.r. | [66] |
| 15 | 2021 | 1 | Asia | Thailand | Spain | A | N.r. | [66] |
| 16 | 2021 | 1 | Asia | India | Spain | A | *Coxiella burnetti* | [66] |
| 17 | 2022 | 1 | Africa | Benin | Turkey | A+ | N.r. | [28] |

*B. burgdorferi* s.l., *Borrelia burgdorferi* sensu lato; EAEC, *Enteroaggregative Escherichia coli*; N.r., none/not reported; Ref., reference; USA, United States of America.

* A+, diagnosed by PCR, culture and/or immunostaining of biopsy/autopsy tissue; A, diagnosed by paired IgG IFA serology; B+, diagnosed by microscopy of blood smear or buffy coat preparation; B, diagnosed by single IgG IFA or ELISA serology; C, diagnosed by IgM serology.

## Outcome

Data on the outcome was available for 395 of the HGA CRID: 12 (3.0%) died due to acute complications related to their infection (Table 21). Of the survivors with respectively available data, 8 (2.1%) were reported to suffer from sequelae (Table 22). The outcome of HGA according to geographic region is shown in Table 23.

## Discussion

### Publications on HGA

Since its discovery in 1994, the number of published case studies on HGA has gradually increased, with more than half of all studies published in the last decade (Fig 6). This increase is likely attributable to increased recognition and awareness among physicians as well as increased availability of specific diagnostic tools. However, the peak in case numbers published

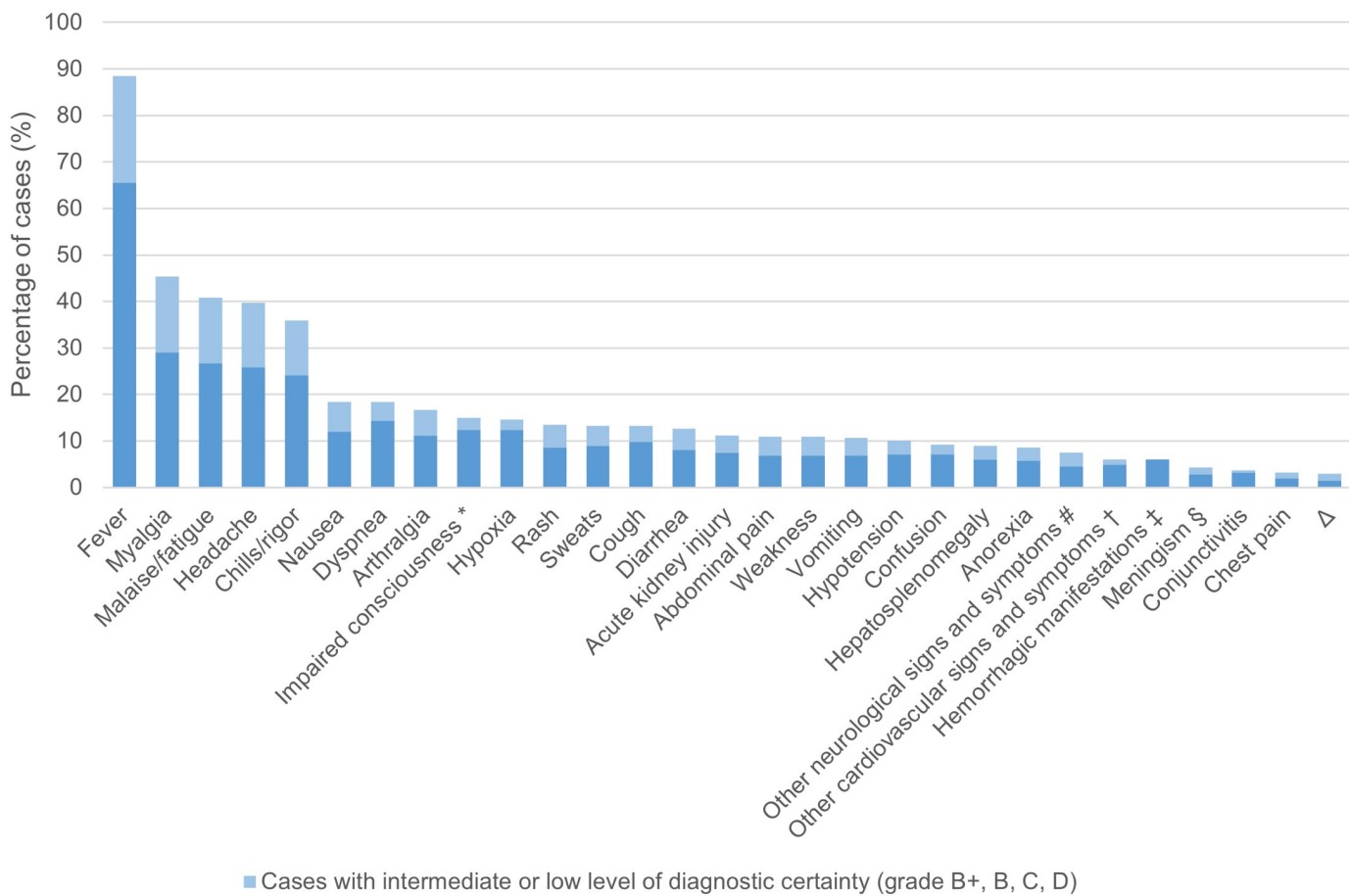

**Fig 7. Signs and symptoms of cases of human granulocytotropic anaplasmosis monoinfection reported with individual data (n = 348 cases).** A+, diagnosed by PCR, culture and/or immunostaining of biopsy/autopsy tissue; A, diagnosed by paired IgG IFA serology; B+, diagnosed by microscopy of blood smear or buffy coat preparation; B, diagnosed by single IgG IFA or ELISA serology; C, diagnosed by IgM serology; D, clinically diagnosed. * definition: altered mental state, confusion, somnolence, delirium, coma. # other neurologic signs and symptoms (cases of all diagnostic certainties): ataxia (11.5%); unilateral Bell's palsy (11.5%); dysarthria (7.7%); severe, lancinating bilateral headache in the distribution of the fifth cranial nerve (3.8%); paresis of arm and hand (3.8%); diplopic images (3.8%); tremor (3.8%); bilateral sensorineural hearing loss (3.8%); relative afferent pupillary defect, dyschromatopsia + visual field defect (3.8%); progressive ascending weakness + sensory disturbances (3.8%); hypesthesia, hyporeflexia + paresthesia of the lower extremities (3.8%); hyperesthesia + radicular pain (3.8%); one-sided paresthesia, pain + weakness of the arm (3.8%); hemispatial inattention, one-sided weakness + hemianopia (3.8%); dysarthria, truncal ataxia, bilateral facial numbness, complete ophthalmoplegia, left ptosis, right facial nerve paresis, symmetrical generalized weakness + hyporeflexia of extremities (3.8%); opsoclonus-myoclonus-ataxia + horizontal hypometric saccades (3.8%); dysphagia (3.8%); expressive aphasia + amnesia (3.8%); tinnitus + unilateral paresthesia of the arm and lip (3.8%); dysarthria + dysphagia + unilateral Bell's palsy (3.8%), seizure (3.8%). † other cardiovascular signs and symptoms (cases of all diagnostic certainties): atrial fibrillation (38.1%); heart failure (38.1%); murmurs (19.0%); atrial fibrillation + heart failure (4.8%). ‡ hemorrhagic manifestations (cases of all diagnostic certainties): petechiae (28.5%); gastrointestinal bleeding (14.2%); hemoptysis (9.5%); hematuria (9.5%); conjunctival bleeding (9.5%); ecchymosis (4.8%); gingival bleeding (4.8%); epistaxis (4.8%); petechiae + gingival bleeding (4.8%); internal + mucosal bleeding (4.8%); unspecified hemorrhagic manifestation (4.8%). § definition: headache plus neck stiffness/meningism and/or photophobia. Δ more rare signs and symptoms not included in the figure (cases of all diagnostic certainties): 2–3%: dizziness, lymphadenopathy, sore throat, unspecified falls; 1–1.9%: vertigo.

in 2011 was due to a large Belgian publication in which 1350 patients presented with suspected tick-borne infection, of whom 418 were diagnosed with HGA [83].

## Epidemiology

Thanks to its decades of case surveillance, the US is the only country for which reliable data on the distribution and incidence of HGA is available, whereas comparatively few data is available

**Table 14. Signs and symptoms of human granulocytotropic anaplasmosis (with A+ and A diagnostic certainty) according to geographic region.**

| Signs / Symptoms | Geographic region | | | | | p-value | | |
|---|---|---|---|---|---|---|---|---|
| | North America [n = 156] % (n) | Europe [n = 46] % (n) | Asia [n = 40] % (n) | Africa [n = 1] % (n) | South America [n = 0] % (n) | North America vs. Europe | North America vs. Asia | Europe vs. Asia |
| Fever | 91 (142) | 89 (41) | 78 (31) | 100 (1) | - | 0.92 | **0.036** | 0.24 |
| Malaise/fatigue | 44 (69) | 43 (20) | 10 (4) | - | - | >0.99 | **<0.001** | **<0.001** |
| Weakness | 12 (18) | 4 (2) | 10 (4) | - | - | 0.26 | >0.99 | 0.41 |
| Myalgia | 42 (66) | 46 (21) | 35 (14) | - | - | 0.82 | 0.51 | 0.43 |
| Arthralgia | 10 (15) | 35 (16) | 8 (3) | 100 (1) | - | **<0.001** | >0.99 | **0.003** |
| Headache | 35 (54) | 61 (28) | 20 (8) | - | - | **0.003** | 0.11 | **<0.001** |
| Rash | 9 (14) | 24 (11) | 10 (4) | 100 (1) | - | **0.024** | 0.77 | 0.15 |
| Gastrointestinal complaints | 30 (47) | 33 (15) | 40 (16) | - | - | 0.89 | 0.32 | 0.63 |
| Anorexia | 9 (14) | 4 (2) | 10 (4) | - | - | 0.53 | 0.77 | 0.41 |
| Nausea | 15 (23) | 20 (9) | 25 (10) | - | - | 0.58 | 0.19 | 0.73 |
| Vomiting | 10 (16) | 13 (6) | 5 (2) | - | - | 0.79 | 0.54 | 0.28 |
| Diarrhea | 12 (19) | 7 (3) | 15 (6) | - | - | 0.42 | 0.83 | 0.29 |
| Abdominal pain | 9 (14) | 9 (4) | 15 (6) | - | - | >0.99 | 0.32 | 0.5 |
| Respiratory symptoms[#] | 30 (47) | 15 (7) | 28 (11) | 100 (1) | - | 0.069 | 0.9 | 0.26 |
| Meningism[†] | 5 (8) | 2 (1) | - | - | - | 0.69 | 0.36 | >0.99 |
| Impaired consciousness[‡] | 37 (24) | 2 (1) | 12 (5) | - | - | **<0.001** | 0.18 | 0.09 |
| Other neurological signs / symptoms[$] | 10 (15) | 7 (3) | - | - | - | 0.77 | **0.044** | 0.24 |
| Cardiovascular signs / symptoms[Δ] | 19 (39) | 4 (2) | 8 (3) | - | - | **0.012** | 0.1 | 0.66 |
| Hepatosplenomegaly | 5 (7) | 22 (10) | 10 (4) | - | - | **<0.001** | 0.24 | 0.16 |

vs., versus.

[#] including: cough, dyspnoe, hypoxia.

[†] definition: headache plus neck stiffness/meningism and/or photophobia.

[‡] definition: altered mental state, confusion, somnolence, delirium, and coma.

[$] including: aphasia, amnesia, ataxia, tremors, dysphagia, dysarthria, palsy, paresis, hearing loss, tinnitus, vision problems, sensory disturbances (numbness, hyper-/hypoesthesia, paresthesia).

[Δ] including: hypotension, atrial fibrillation, heart failure, murmurs.

for the rest of the world. The epidemiology of HGA within the USA as well as worldwide is defined by the distribution of the transmitting *Ixodes* spp. tick vectors with the vast majority of cases being reported from the U.S. (primarily from the Northeast and Upper Midwest [5]), Europe, and Asia (Table 5). Because some other hard-tick species are also capable of transmitting HGA, the disease also occurs in other parts of the world. It is important to acknowledge that Fig 2 is not so much showing the true global epidemiological picture of HGA, but rather the regions where the respective diagnostic capacity and expertise is available. In particular, very little is known about HGA in South America and Africa and it is of note that up to date, not a single PCR-confirmed HGA case has been reported from Central or South America. The presence of *A. phagocytophilum* in South Africa is supported by a single study reporting PCR-positive human blood samples from patients with non-malarial fevers as well as PCR-positive

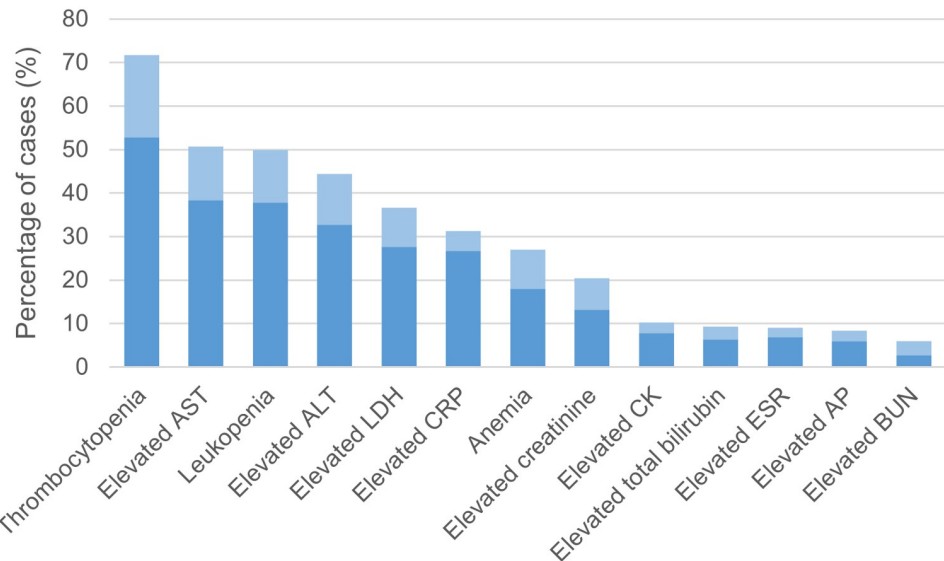

**Fig 8. Laboratory findings in human granulocytotropic anaplasmosis (n = 333 cases).** A+, diagnosed by PCR, culture and/or immunostaining of biopsy/autopsy tissue; A, diagnosed by paired IgG IFA serology; ALT, alanine aminotransferase; AP, alkaline phosphatase; AST, aspartate aminotransferase; B+, diagnosed by microscopy of blood smear or buffy coat preparation; B, diagnosed by single IgG IFA or ELISA serology; BUN, blood urea nitrogen; C, diagnosed by IgM serology; CK, creatine kinase; CRP, C-reactive protein; D, clinically diagnosed; ESR, erythrocyte sedimentation rate; LDH, lactate dehydrogenase. [The cut-offs defining abnormal values of the individual laboratory parameters are listed in S6 Text].

**Table 15. Medians and ranges of laboratory parameters in human granulocytotropic anaplasmosis (with A+ and A diagnostic certainty).**

| Laboratory parameters | Median | Range | Number of cases with available data (n) |
|---|---|---|---|
| Leukocytes ($10^3$/μl) | 3.4 | 0.4–28.5 | 163 |
| Thrombocytes ($10^3$/μl) | 63 | 3–323 | 171 |
| Hemoglobin (g/dl) | 11.2 | 5.2–15.4 | 65 |
| Hematocrit (%) | 35 | 21–44 | 18 |
| AST (U/l) | 98 | 10–1049 | 135 |
| ALT (U/l) | 71 | 8–2471 | 126 |
| AP (U/l) | 136 | 22–1200 | 30 |
| Total bilirubin (μmol/l) | 22 | 5–253 | 29 |
| CRP (mg/l) | 110 | 0–380 | 89 |
| ESR (mm/h) | 32 | 3–94 | 32 |
| LDH (U/l) | 468 | 140–6592 | 87 |
| BUN (mg/dl) | 24 | 17–117 | 15 |
| Creatinine (μmol/l) | 141 | 44–8889 | 45 |
| CK (U/l) | 255 | 22–57624 | 41 |

ALT, alanine aminotransferase; AP, alkaline phosphatase; AST, aspartate aminotransferase; BUN, blood urea nitrogen; CK, creatine kinase; CRP, C-reactive protein; ESR, erythrocyte sedimentation rate; LDH, lactate dehydrogenase.

**Table 16. Abnormal laboratory findings in human granulocytotropic anaplasmosis (with A+ and A diagnostic certainty) according to geographic region.**

| Abnormal laboratory finding | | Geographic region | | | | | p-value | | |
|---|---|---|---|---|---|---|---|---|---|
| | | North America [n = 145] % or median [IQR] (n) | Europe [n = 46] % or median [IQR] (n) | Asia [n = 39] % or median [IQR] (n) | Africa [n = 1] % or median [IQR] (n) | South America [n = 0] % or median [IQR] (n) | North America vs. Europe | North America vs. Asia | Europe vs. Asia |
| Thrombocytopenia | | 82 (119) | 57 (26) | 79 (31) | - | - | **<0.001** | 0.891 | **0.044** |
| | PLT ($10^3$/μl) | 51 [25–99] (102) | 82 [51–180] (34) | 63 [30–112] (35) | - | - | **0.005** | 0.558 | 0.069 |
| Leukopenia | | 54 (79) | 50 (23) | 62 (24) | - | - | 0.717 | 0.544 | 0.396 |
| | WBC ($10^3$/μl) | 3.5 [2.5–5.1] (92) | 3.4 [2.3–6.1] (35) | 2.9 [2.1–4.4] (35) | 20.0 [N.c.] (1) | - | 0.95 | 0.122 | 0.237 |
| Anemia | | 32 (47) | 9 (4) | 21 (8) | 100 (1) | - | **0.001** | 0.213 | 0.133 |
| | Hb (g/dl) | 11.2 [10.0–13.0] (53) | 10.3 [9.1–12.1] (6) | 12.6 [8.8–13.2] (5) | 12.7 [N.c.] (1) | - | 0.436 | 0.857 | 0.854 |
| Elevated liver enzymes* | | 66 (95) | 72 (33) | 85 (33) | - | - | 0.547 | **0.035** | 0.246 |
| Elevated ALT | | 38 (55) | 65 (30) | 62 (24) | - | - | **0.002** | **0.013** | 0.9 |
| | ALT (U/l) | 76 [49–121] (62) | 105 [52–164] (32) | 53 [35–78] (31) | 24 [N.c.] (1) | - | 0.196 | 0.054 | **0.009** |
| Elevated AST | | 48 (69) | 61 (28) | 79 (31) | - | - | 0.161 | **<0.001** | 0.105 |
| | AST (U/l) | 100 [57–172] (72) | 107 [53–151] (31) | 87 [58–113] (31) | 20 [N.c.] (1) | - | 0.684 | 0.199 | 0.371 |
| Elevated creatinine or BUN* | | 28 (41) | - | 5 (2) | - | - | **<0.001** | **0.001** | 0.207 |
| Acute kidney injury* | | 28 (41) | - | 3 (1) | - | - | **<0.001** | **0.018** | 0.458 |
| Elevated inflammation markers (CRP or ESR) | | 20 (29) | 65 (30) | 74 (29) | 100 (1) | - | **<0.001** | **<0.001** | 0.499 |
| | CRP (mg/l) | 126 [90–185] (27) | 92 [52–155] (30) | 85 [47–163] (29) | 150 [N.c.] (1) | - | 0.07 | 0.119 | 0.867 |
| Elevated LDH | | 30 (43) | 65 (30) | 49 (19) | - | - | **<0.001** | 0.04 | 0.188 |
| Elevated CK | | 8 (11) | 2 (1) | 33 (13) | - | - | 0.299 | **<0.001** | **0.0001** |

ALT, alanine aminotransferase; AST, aspartate aminotransferase; BUN, blood urea nitrogen; CK, creatine kinase; CRP, C-reactive protein; ESR, erythrocyte sedimentation rate; IQR, interquartile range; LDH, lactate dehydrogenase; N.c., not calculated; vs., versus; PLT, platelet/thrombocyte count; WBC, white blood cell count.

* according to the authors.

samples from rodents, dogs and cattle [84]. The only other evidence for the presence of HGA in sub-Saharan Africa comes from three travellers who were diagnosed with travel-associated HGA infection after their return from Africa (Table 13). *A. phagocytophyilum* has also been reported in ticks (*Rhiphicephalus* and *Amblyomma* spp.) and animals investigated in several African countries (Zambia, Zimbabwe, Tunesia), but it is unclear whether those reports were truly *A. phagocytophilum* and not other *Anaplasma* species, as the 16S rRNA segment used for amplification was short and thus possibly inadequate for species discrimination [84]. More data on tick vectors and animal reservoirs of *A. phagocytophilum* can be found in the respective systematic review and meta-analysis by Karshima et al. [85,86].

## Non-*phagocytophilum* Anaplasmosis

Although considerably less common overall, some HA cases caused by *Anaplasma* spp. other than *A. phagocytophilum* have been reported (Table 6). For some cases infected with *A. bovis*, *A. capra*, or *A. ovis*, a positive serology for *A. phagocytophilum* was reported, suggesting some

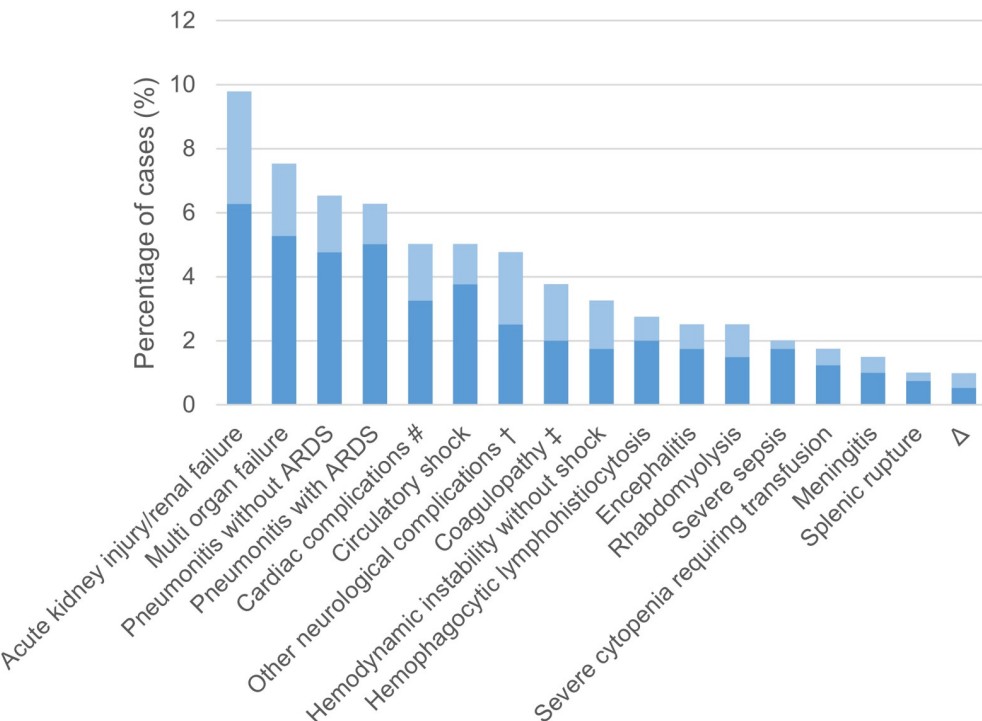

**Fig 9. Frequency of complications in HGA monoinfection cases reported with individual data (n = 398).** * ARDS, acute respiratory distress syndrome; A+, diagnosed by PCR, culture and/or immunostaining of biopsy/autopsy tissue; A, diagnosed by paired IgG IFA serology; ARDS, acute respiratory distress syndrome; B+, diagnosed by microscopy of blood smear or buffy coat preparation; B, diagnosed by single IgG IFA or ELISA serology; C, diagnosed by IgM serology; D, clinically diagnosed. * as in many cases, multiple complications were concomitantly present, the number of complications exceeds the number of cases. # cardiac complications: cardiomyopathy and/or heart failure (45.0%); cardiac arrhythmia (35.0%); carditis (15.0%); heart failure + cardiac arrhythmia (5.0%). † other neurologic complications: cranial neuritis (36.8%); other neuritis or myelitis (36.8%); acute inflammatory demyelinating polyradiculoneuritis) (10.5%); hemiparesis (5.3%); seizure (5.3%); unspecified focal neurological symptoms (5.3%). ‡ coagulopathy: unspecified coagulopathy (40.0%); disseminated intravascular coagulopathy (26.7%); coagulopathy with clinical bleeding (13.3%); thrombotic thrombocytopenic purpura-like symptoms (13.3%); multiple pulmonary emboli (6.6%). Δ more rare complications not included in the figure: <1%: hemorrhagic manifestations requiring transfusion, pulmonary edema, 2ndary opportunistic infections (pulmonary aspergillosis or pulmonary cryptococcosis), syncope, peritonitis, hyponatremia, miscarriage, pericardial tamponade, severe polyarthritis, Sweet syndrome, Kawasaki syndrome, nephrotic syndrome, orchitis, autoimmune hemolytic anemia, urticaria + angioedema.

**Table 17. Occurrence of complications in human granulocytotropic anaplasmosis (with A+ and A diagnostic certainty) according to geographic region.**

| Occurrence of complications in HGA cases | Geographic region | | | | | p-value | | |
|---|---|---|---|---|---|---|---|---|
| | North America [n = 171] % (n) | Europe [n = 50] % (n) | Asia [n = 39] % (n) | Africa [n = 1] % (n) | South America [n = 0] % (n) | North America vs. Europe | North America vs. Asia | Europe vs. Asia |
| Complications | 56 (95) | 24 (12) | 23 (9) | 100 (1) | - | < 0.001 | < 0.001 | 1 |
| No complications | 44 (76) | 76 (38) | 77 (30) | - | - | | | |

HGA, human granulocytotropic anaplasmosis; vs., versus.

**Table 18. Reported antimicrobial treatment of human granulocytotropic anaplasmosis (n = 390) considered effective\*.**

| Antimicrobial treatment | Number of cases n (%) |
|---|---|
| Doxycycline | 361 (92.6) |
| Minocycline | 14 (3.6) |
| Rifampicin | 10 (2.6) |
| Chloramphenicol | 1 (<1.0) |
| Multiple compounds | 4 (1.0) |
| Rifampicin → doxycycline | 1 |
| Doxycycline → rifampicin | 1 |
| Doxycycline → + chloramphenicol | 1 |
| Minocycline i.v. → doxycycline p.o. | 1 |

→, consecutive administration; → +, treatment escalation;

i.v., intravenous; p.o., per oral.

\* tetracyclines (doxycycline, minocycline) and rifampicin were considered effective/appropriate; chloramphenicol was also considered effective/appropriate, although evidence is limited; beta-lactams, quinolones, macrolides, aminoglycosides, glycopeptides, nitroimidazoles, sulfonamides, and lincosamide were considered ineffective/inappropriate.

degree of serological cross-reactivity of assays [7,11,12]. *Anaplasma platys*, the causative agent of canine cyclic thrombocytopenia has been detected in dogs on almost all continents [87,88] but up to date, reports of human infections are restricted to Central and South America (Table 6). No serological assay for detecting human *Anaplasma platys* infections is available, and the IFA used to diagnose the infection in dogs is reported to not detect human infection [23]. *Anaplasma bovis*, known for causing infections in cattle, is well established in Asia and Africa [89] but to date, reports of human infections are restricted to China (Table 6). Recently, an *A. bovis*-like agent infecting four humans in the US was reported [90]. Human infections with *Anaplasma capra*, a species first described in goats in China in 2015, later also in animals in other Asian countries and recently in France [91], have to date only been reported from China [11]. Reported human infections with *Anaplasma ovis*, known for infecting erythrocytes of goats in tropical and subtropical regions are limited to one case from Cyprus and one case from Iran (Table 6).

Recently, a HGA case due to *Candidatus* Anaplasma sparouinense, an isolate closely related to Brazilian wildlife isolates, was reported in French Guiana. The patient, a 58-year-old man with asplenia, was diagnosed with anaplasmosis by PCR in 2021. Interestingly, retrospective

**Table 19. Appropriate antimicrobial treatment of human granulocytotropic anaplasmosis according to geographic region.**

| Antimicrobial compound used to treat HGA | Geographic region | | | | | p-value | | |
|---|---|---|---|---|---|---|---|---|
| | North America [n = 254] % (n) | Europe [n = 95] % (n) | Asia [n = 48] % (n) | Africa [n = 3] % (n) | South America [n = 1] % (n) | North America vs. Europe | North America vs. Asia | Europe vs. Asia |
| Appropriate | 93 (236) | 55 (52) | 83 (40) | 33 (1) | 100 (1) | < **0.001** | 0.058 | **0.001** |
| Inappropriate | 7 (18) | 45 (43) | 17 (8) | 67 (2) | - | | | |

HGA, human granulocytotropic anaplasmosis; vs., versus.

**Table 20. Time windows elapsing between onset of symptoms and presentation to a medical facility and between presentation to a medical facility and first administration of appropriate antimicrobial treatment.**

| | All HGA cases median (IQR) | North America median (IQR) | Europa median (IQR) | Asia median (IQR) | p-value | | |
|---|---|---|---|---|---|---|---|
| | | | | | North America vs. Europe | North America vs. Asia | Europe vs. Asia |
| Time window between onset of symptoms and presentation to a medical facility in days | [n = 217] 5 (3–7) | [n = 126] 5 (3–7) | [n = 67] 4 (3–8) | [n = 23] 6 (4–9) | 0.873 | 0.54 | 0.482 |
| Time window between presentation to a medical facility and first administration of appropriate antimicrobial treatment in days | [n = 81] 3 (0–5) | [n = 55] 3 (1–7) | [n = 18] 3 (0–5) | [n = 8] 2 (1–3) | 0.246 | 0.283 | 0.909 |
| Time window between onset of symptoms and first administration of appropriate antimicrobial treatment in days | [n = 112] 6 (3–10) | [n = 70] 6 (3–14) | [n = 18] 4 (3–7) | [n = 24] 6 (5–8) | 0.271 | 0.91 | 0.187 |

HGA, human granulocytotropic anaplasmosis; IQR, interquartile range; vs., versus.

Note: data from Africa and South America are absent in the table, as no such data have been reported.

Table 21. Reported fatal cases of human granulocytotropic anaplasmosis (n = 12).

| No. | Year of publication | Age of patient (years) | Sex of patient | Country of infection | Pre-existing medical condition | Immunosuppressive treatment | Hospital admission | Time between first symptoms and medical presentation (days) | Level of diagnostic certainty# | Antimicrobial treatment | Time between presentation to hospital and specific therapy (days) | Time from first symptoms to death (days) | Complications/cause of death | Ref. |
|---|---|---|---|---|---|---|---|---|---|---|---|---|---|---|
| 1 | 1994 | 80 | Male | USA | CLL with Richter's syndrome, splenectomy | High-dose corticosteroid | Yes | N.r. | A+ | Doxycycline, erythromycin | N.r. | 14 | Multi-organ failure, gastroesophageal hemorrhage, severe HSV1 oesophagitis, pulmonary cryptococcosis | [35,67] |
| 2 | 1994 | 80 | Male | USA | N.r. | N.r. | Yes | 4 | A+ | Gentamicin, ceftriaxone, metronidazole, clindamycin | N.r. | N.r. | Multi-organ failure (ARDS, anuric renal failure), gastroesoph. hemorrhage due to severe Candida oesophagitis | [35,67] |
| 3 | 1995 | 71 | Male | USA | COPD, emphysema, coronary bypass graft, nephrectomized, hemicolectomized due to diverticulitis | N.r. | Yes | 4 | B+ | Doxycycline | 5 | 28 | Acute myocardial infarction, heart failure, pulmonary aspergillosis, coagulopathy, clinical bleeding | [68] |
| 4 | 1998 | 44 | Male | USA | N.r. | N.r. | No | 7 | A+ | Amoxicillin | N.r. | 12 | Pancarditis | [69] |
| 5 | 2000 | 9 | Male | USA | N.r. | N.r. | Yes | N.r. | B | N.r. | N.r. | N.r. | N.r. | [70] |
| 6 | 2012 | 64 | Male | USA | Bone marrow transplantation due to leukemia | Immunosuppressant medication after bone marrow transplant | Yes | 21 | A+ | Doxycycline, "broad-spectrum antibiotics" | <1 | 28 | Hemodynamic instability, respiratory failure | [71] |
| 7 | 2017 | 32 | Male | South Korea | N.r. | N.r. | Yes | 4 | A+ | N.r. | N.r. | 12 | HLH | [72] |
| 8 | 2017 | 53 | Male | South Korea | N.r. | N.r. | Yes | 1 | A+ | N.r. | N.r. | 30 | HLH | [72] |
| 9 | 2018 | 78 | Male | USA | Coronary artery disease, heart failure, type 2 diabetes, chronic kidney disease | N.r. | Yes | N.r. | A+ | Doxycycline, vancomycin, piperacillin tazobactam | N.r. | 5 | Multi-organ failure (respiratory failure, refractory circulatory shock) | [16] |
| 10 | 2018 | 68 | Female | USA | Complex medical history including heart failure | N.r. | Yes | 7 | B+ | Doxycycline, ceftriaxone, metronidazole | 10 | N.r. | Multi-organ failure (acute kidney injury, circulatory shock, splenic rupture) | [73] |
| 11 | 2020 | 55 | Male | USA | Adrenal insufficiency due to autoimmune polyendocrine syndrome type II | N.r. | Yes | N.r. | B+ | Doxycycline, cefepime | N.r. | N.r. | Multi-organ failure (circulatory shock, respiratory failure, heart failure, encephalopathy) | [74] |
| 12 | 2022 | 89 | Male | USA | Multiple comorbidities (end-stage renal disease, hemodialysis, heart failure) | N.r. | Yes | 7 | A+ | Doxycycline, "broad-spectrum antibiotics" | N.r. | 16 | Multi-organ failure (circulatory shock, ARDS, encephalopathy) | [75] |

ARDS, acute respiratory distress syndrome; CLL, chronic lymphocytic leukemia; COPD, chronic obstructive pulmonary disease; HLH, hemophagocytic lymphohistiocytosis; HSV, herpes simplex virus; N.r., none/not reported; Ref, reference; USA, United States of America.

# A+, diagnosed by PCR, culture and/or immunostaining of biopsy/autopsy tissue; B+, diagnosed by microscopy of blood smear or buffy coat preparation; B, diagnosed by single IgG IFA or ELISA serology.

**Table 22. Reported sequelae of human granulocytotropic anaplasmosis (n = 8).**

| No. | Year of publication | Age of patient (years) | Sex of patient | Country of infection | Pre-existing medical condition | Immuno-suppressive treatment | Time between first symptoms and medical presentation (days) | Level of diagnostic certainty* | Antimicrobial therapy | Time between presentation to hospital and specific therapy (days) | Complications | Sequelae | Ref. |
|---|---|---|---|---|---|---|---|---|---|---|---|---|---|
| 1 | 1996 | 41 | Male | USA | N.r. | N.r. | 12 | B | Doxycycline, cefoperazone, erythromycin, cefaclor | 140 | Neuritis (Brachial plexopathy leading to *scapula alata*) | Brachial plexopathy with gradual improvement after 9 months | [76] |
| 2 | 1998 | 48 | Female | USA | N.r. | N.r. | 4 | A+ | Doxycycline | 0 | Acute inflammatory demyelinating polyneuropathy | Persisting neurological deficits 6 months after acute event | [77] |
| 3 | 2000 | 42 | Female | USA | N.r. | N.r. | 7 | A+ | Doxycycline, ceftriaxone | N.r. | Cranial nerve neuritis (facial diplegia) | Gradual improvement but still slight facial muscle weakness and mild dysarthria at 5 months | [33] |
| 4 | 2003 | 41 | Female | USA | N.r. | N.r. | N.r. | A | Doxycycline | 12 | Optic neuritis | Gradual improvement but residual paracentral scotoma at 2 months | [78] |
| 5 | 2014 | 78 | Male | USA | Chronic kidney diseases, chronic heart disease, interstitial lung disease | N.r. | 3 | A+ | Doxycycline, "broad-spectrum antibiotics" | N.r. | Multi-organ failure (ARDS, ARF requiring dialysis, cardiomyopathy) | Persisting renal failure requiring dialysis | [79] |
| 6 | 2019 | 78 | Male | USA | History of non-specified vasculitis | N.r. | N.r. | A+ | Doxycycline | N.r. | Diffuse alveolar hemorrhage with respiratory failure, septic shock, ARF requiring dialysis | Renal failure requiring dialysis (unclear whether temporary or permanent) | [80] |
| 7 | 2020 | 83 | Male | USA | Chronic kidney disease, coronary artery disease (coronary bypass graft) | N.r. | 5 | A+ | Doxycycline | 5 | ARF requiring dialysis, coagulopathy | Renal failure requiring dialysis (unclear whether temporary or permanent) | [81] |
| 8 | 2021 | 75 | Male | Belgium | CLL | N.r. | 10 | A | Doxycycline, ceftriaxone | N.r. | Neuritis (acute brachial plexopathy) | Overall improvement but persisting residual weakness of the hand at 1 year | [82] |

ARDS, acute respiratory distress syndrome; ARF, acute renal failure; CLL, chronic lymphatic leukemia; N.r., none/not reported; Ref., reference; USA, United States of America.
* A+, diagnosed by PCR, culture and/or immunostaining of biopsy/autopsy tissue; A, diagnosed by paired IgG IFA serology; B, diagnosed by single IgG IFA or ELISA serology.

**Table 23. Outcome of human granulocytotropic anaplasmosis (with A+ and A diagnostic certainty) according to geographic region.**

| Outcome of HGA | Geographic region | | | | | p-value | | |
|---|---|---|---|---|---|---|---|---|
| | North America [n = 170] % (n) | Europe [n = 49] % (n) | Asia [n = 39] % (n) | Africa [n = 1] % (n) | South America [n = 0] % (n) | North America vs. Europe | North America vs. Asia | Europe vs. Asia |
| Survival | 96 (164) | 100 (49) | 95 (37) | 100 (1) | - | 0.342 | 0.644 | 0.193 |
| Death | 4 (6) | 0 (0) | 5 (2) | - | - | | | |

HGA, human granulocytotropic anaplasmosis; vs., versus.

analysis of an earlier blood sample from 2019 (when the patient was reported as asymptomatic) was already PCR-positive and microscopically positive for inclusion bodies. However, it is unclear whether his symptoms in 2021 were caused by *Anaplasma* or by a serologically also suspected *Coxiella burnetii* infection, and whether the long persistence of *Anaplasma* may have been influenced by the fact that he had been splenectomized in the past [13].

## Coinfections

Among the reviewed HGA cases, we found a reported coinfection rate with other tick-borne pathogens of 15.8% (465/2942; Table 7). However, reported data on the rate of co-infection in HGA cases should be interpreted with caution, as sample sizes and representativeness often limits generalizability. In addition, tests for coinfections are often neither uniformly nor systematically carried out, very often limited to serology and the interpretation by the authors is frequently not well described. Thus, the probability of correctly diagnosed co-infections must generally be viewed critically. If a pathogen has a regionally high prevalence and incidence, co-infection is plausible, even if the transmission route is different. However, if a pathogen is rare and the route of transmission not identical, the plausibility of co-infection is considerably lower. This is especially true if the level of diagnostic certainty of the applied tests is low.

Although coinfections do occur, and testing for coendemic pathogens sharing identical tick vectors should be considered, the coinfection rate found in our analysis is very likely an overestimate as most studies directly examining this in a systematic and unbiased fashion find a coinfection rate of less than 10%. Fortunately, most tick-borne bacterial infections respond to doxycycline, so undetected co-infections are also covered, although recommendations for the duration of antimicrobial treatment may vary depending on the pathogen. The data was too sparse and heterogeneous to allow a valid assessment of a putative difference in disease severity between cases with mono- and co-infections. We therefore refrained from attempting a corresponding analysis. The few studies which investigated the question whether HGA co-infections may be associated with more severe symptoms than HGA mono-infections focus on Lyme co-infection and came to conflicting conclusions [92–94]. It appears that these conflicting results accrue from, or are at least partly attributable to, different case definitions and selection/referral bias [95].

Cases of coinfections with TBEV are, in line with the geographical distribution, only reported from Europe and Asia and more common among CRNID. The latter is attributable to the fact that these cases were primarily reported from cohort data of patients with tick-borne encephalitis [96–98].

In three CRID, coinfection of *A. phagocytophilum* and *E. chaffeensis* was postulated. In two of those cases, both diseases were diagnosed only serologically [99,100], while in one case, PCR

for both agents was positive (with paired serology being negative for both agents) [101]. As cross reactivity in serological assays for *A. phagocytophilum* and *E. chaffeensis* is not uncommon, results should be interpreted with caution [102–105]. The US CDC suggests that in the event that serology is positive for both pathogens, the stronger antibody response should generally be considered to be the one directed against the actual causative pathogen [20]. Unfortunately, no titres were reported in the two solely serologically diagnosed coinfections of *A. phagocytophilum* and *E. chaffeensis*. In the case with positive PCR for both agents, the used assay was not specified. Since no assay will be 100% specific, the possibility of a false-positive result in this case has also to be anticipated. Since no vector is yet known to transmit both pathogens, the probability of a coinfection with both pathogens is overall very low.

In 15 CRID, coinfection with *Orientia tsutsugamushi* was reported. All these cases were reported in context of two Asian case series of confirmed scrub typhus in which the patients were additionally screened for *A. phagocytophilum* coinfection. HGA coinfection was reported in 8/274 (2.9%) and 7/167 (4.2%) of scrub typhus cases, respectively [106,107]. Although *O. tsutsugamushi* is transmitted exclusively by larvae of thromiculid mites, the high co-infection rate may be conclusively explained by the overall high incidence of scrub typhus in endemic areas and the high risk of exposure to both vectors [108].

With regard to the reported co-infection cases of HGA with *Bartonella henselae*, it should be noted that although *Bartonella* spp. are not generally considered tick-borne pathogens, there is some evidence that they can be transmitted by ticks [109–112].

## Diagnostic

The most frequently reported diagnostic methods used to diagnose HGA are PCR and serology (Table 8). Regarding serology, the investigation of paired samples remains the gold standard. In our analysis, comparatively few cases were diagnosed this way. This may be explained by the fact that suspected cases will primarily be empirically treated as the serological result of paired samples takes far too long to be helpful for making an acute therapeutic decision and the effort to retrospectively confirm the diagnosis in a successfully treated patient may not be considered worthwhile. In addition, PCR is increasingly available and cases diagnosed by PCR will unlikely in parallel be tested and followed up by serology. However, if availability permits, confirmation with both methods should be considered to rule out false positive results by one or the other method.

If single titre serology is used, a IgG cut-off of 1:64 is seen as supporting the diagnosis [17]. However, as antibodies against *A. phagocytophilum* can persist for several years after acute infection and seropositivity rates may reach up to 37% in some populations [22,104], some authors recommend using a higher cut-off value to increase specificity and better discriminate acute infection from background seropositivity [17,113].

Culture was rarely performed, as it is technically challenging as well as resource and time demaning. If performed, culture was mostly done in HL-60 cell line. Of note is one rather exceptional report where the patient's own leukemic cells were used to culture *A. phagocytophilum* [32].

Diagnostic biopsy was only reported in a few cases. Bone marrow was the most frequently examined tissue, with morulae regularly found within granulocytic precursor cells. In general however, *A. phagocytophilum* seems to have a higher affinity for more mature granulocytes and is therefore reported to be more reliably found in peripheral blood than in bone marrow [114]. When the presence of *A. phagocytophilum* is detected by immunhistochemistry (IHC) staining of samples from parenchymal organs, such as the kidney, placenta or spleen, the

pathogen is always detected within cells of the myeloid cell lineage and never in the tissue cells of the organ itself [67,69,115].

The microscopic detection of morulae in blood smears or buffy coat preparations was only the third most common diagnostic method used to establish the diagnosis after PCR and serology. Interestingly, we found considerable differences regarding the reported frequency of microscopically detected morulae among PCR-confirmed HGA cases between different geographic regions (Table 9). Whether this is due to biological differences of the pathogen, or the differences accrue from the variance of regional microscopic expertise, the communication of the suspected diagnosis to the laboratory staff or the sensitization of laboratory staff in endemic areas (influencing the specific watch out for morulae and the number of granulocytes routinely looked at) is not answerable. Regarding the latter it is of note, that buffy coat preparations provide a higher sensitivity compared to blood smears, as only a fraction of circulating blood cells contain morulae [116]. Our analysis also included cases with confirmed HGA showing morulae in other cell lines than granulocytes, which underlines that the tropism of *A. phagocytophilum* for a particular leukocyte type is not 100%.

## Patients' characteristics

HGA was more commonly diagnosed in males than in females (M:F ratio = 1.44) and mostly seen in middle-aged and older individuals, which corresponds to US surveillance data [5,19]. In Europe and Asia, the sex difference was less pronounced than in North America, with infection in females being more common than in males in Asia. In addition, cases from Europe were on average younger than cases from North America and Asia. We speculate that these differences accrue from different patterns of recreational and/or occupational tick exposure. In addition, susceptibility for infection may influence the epidemiological pattern: being immunocompromised/-suppressed was reported in 7.4% of the cases, which is similar to what national US surveillance data reported (4–11.3% [19,117]). We found immunodeficiency/suppression to be statistically significantly more common in cases from North America than in cases from Europe and Asia (Table 11). However, whether this is a true effect or caused by reporting bias remains unclear.

We found an overall hospitalization rate of 59.2%, which is considerably higher than what has been reported in the US case surveillance, where hospitalization rates of 36% (687/1907; from 2000–2007) and 30.8% (1827/5937; from 2008–2012) were reported [4,19]. We consider the almost double as high hospitalization rate found in our analysis to accrue from reporting/ publication bias and not reflect the true picture.

## Route of transmission

The most common route of transmission of HGA is by tick bite, which is confirmed by the fact that in 95.4% of the reviewed HGA cases the suspected route of transmission was a tick bite. Studies suggest that *A. phagocytophilum* can already be transmitted when a tick is attached for less than 24 hours which is considerable faster than what is reported for e.g. *B. burgdorferi* [118].

Eleven transfusion-transmitted cases have been reported from America and Europe, with transfusions of red blood cells and platelets being the culprit blood product (Table 11). In some cases, infection occurred despite storage times of up to 30 days and leukoreduction [40]. One study found that leukoreduction can reduce but not eliminate the risk of infection [119]. This is also evident in the cases we reviewed, in which more than half of the cases had received leukoreduced blood products (Table 11). Of the cases where clinical data from the donors were available, the donor usually had an a- or oligosymptomatic course of disease

[43,45,46,120]. With PCR a highly sensitive screening method for blood products would be available, but considering the overall low incidence and risk of infection, routine screening of blood products for *A. phagocytophilum* is not considered cost-effective and, thus to our knowledge not implemented in any country [120].

In 2008, a Chinese publication postulated potential nosocomial transmission of *A. phagocytophilum* after 9 family members and health care workers who had contact to body fluids of a heavily bleeding patient became ill and were tested positive by PCR [34]. However, in 2012 it was retrospectively clarified, that not *A. phagocytophilum* but *Severe Fever with Thrombocytopenia Syndrome Virus* (SFTSV), a bunyavirus discovered in 2010, was the pathogen causing this cluster of nosocomial cases and the positive PCR result for anaplasmosis was retrospectively questioned [34,121,122].

We found a total of six cases where the authors speculated that the route of transmission was contact to game blood or handling/slaughtering game without suitable protective equipment. This transmission route has not yet been proven for *A. phagocytophilum*, but is well documented for *A. marginale* [123]. However, given the not insignificant risk of tick exposure among hunters and butchers, transmission through an undetected tick bite in these cases could be an alternative explanation [118].

Our analysis included 17 infections during pregnancy (Table 12), including one case of perinatal transmission [38]. All infections showed a favorable outcome for mother and infant. In the case with perinatal transmission, the mother became ill at 39 weeks of pregnancy and was not treated until after birth. Her infant showed first symptoms on the 8th day of life. Whether the infection was intrauterine or intrapartum transmitted or through breastfeeding remains unclear, but intrauterine transplacental transmission, as has been observed in animals, is considered most likely [38,124]. In most cases reported during pregnancy, the course of disease was mild. Miscarriage was reported in only three cases infected during early pregnancy. Two of these women were co-infected with *B. garinii* [53], which may be considered the more likely cause of miscarriage in these cases, as Lyme borreliosis is a reported risk factor for adverse birth outcomes [125]. In the third case, the diagnosis of HGA was purely speculative, based on clinical signs and symptoms only, and no HGA specific diagnostic testing was done [52]. Thus, HGA does not appear to be associated with unfavorable birth outcomes.

## Travel-related cases

There were a few travel-related cases (Table 13), most of them from countries where HGA has been described rarely or not at all. A Spanish single-centre study looking at 141 returning travellers with non-malarial fever reported an overall incidence rate of HGA of 19.9 cases/1000 people with undifferentiated non-malarial fever per week of travel [66]. However, as the diagnostic methods used in some cases were of low diagnostic certainty, the actual incidence might be lower. However, the study highlights the potential sentinel role of international travellers and suggests that HGA may be far more widespread than assumed and may even be an underestimated cause of fever in some regions of the world as well as in travellers returning from these regions [66].

## Signs and symptoms

HGA presents similarly to many other febrile illnesses without specific signs and symptoms (Fig 7) and can be easily misdiagnosed or overlooked because of its often oligosymptomatic and self-limiting course as well as its sensitivity to doxycycline, which is often used as an empiric treatment for tick-borne infections (e.g., Lyme disease). The finding of seroconversion [126–128] and even PCR positivity of asymptomatic individuals in several studies

[43,45,46,120,129–131] reflects that subclinical infection is probably common. The most common symptoms in symptomatic patients include fever, chills, malaise, headache, and myalgia. Some patients also develop pulmonary symptoms such as cough or shortness of breath, gastrointestinal symptoms including nausea, vomiting, abdominal pain and diarrhea, or cardiovascular or neurological signs and symptoms. A rash was reported in 13.5% of monoinfected CRID, 8.1% of CRNID, and 25.9% of CRID with coinfection(s). Most sources put the incidence of skin rash at less than 10% of HGA cases [14,17,35]. Considering that a rash may be rather rare in HGA, some authors emphasize the importance of considering other or additional tick-borne infections in patients presenting with rash after having sustained a tick-bite [3]. The same is true for other symptoms highly suggestive of being caused by another causative pathogen (like e.g. in the case of Bell's palsy or pronounced monoarthrisis where *B. burgdorferi* is more plausible) or symptoms which are rather atypical and, thus questionable in the context of HGA (e.g. cardiomyopathy/ heart failure, disseminated intravascular coagulation). When comparing symptoms of HGA across different geographic regions, we observed some statistically significant differences in our data set (Table 14), suggesting that there may be differences in the clinical presentation of HGA in different geographical regions. Although the interpretation of this data is limited due to the nature of the data analyzed, it will be interesting to see if future studies can confirm the existence of these differences.

## Laboratory findings

Laboratory findings in HGA cases are not specific, but often characterized by the combination of cytopenia (with thrombocytopenia being the most common, followed by leukopenia and less commonly mild non-hemolytic anemia) and elevated transaminase levels. However, this pattern may also be observed with other infections, including many other tick-borne infections (e.g., rickettsiosis, ehrlichiosis, tularemia, babesiosis, tick-borne relapsing fever, Q fever, and tick-borne arboviral infections) [132]. Antimicrobial treatment with doxycycline leads to the resolution of thrombocytopenia and leukopenia within one week, while anemia and the elevation of transaminase levels resolve more slowly [133].

When comparing laboratory findings across different geographic regions, we observed some statistically significant differences in our data set (Table 16), suggesting that there may be differences regarding the laboratory presentation of HGA in different geographical regions. Although the interpretation of this data is limited due to the nature of the data analyzed, it will be interesting to see if future studies can confirm the existence of these differences.

## Complications

The reported rate of complications among the reviewed monoinfected CRID was 40.5% and thus rather high when compared to 3.0–7.8% of cases sustaining a life-threatening complication according to US-case surveillance data [4,19]. This difference is very likely explained by reporting and publication bias, but possibly also by differences in case definition. We defined a complicated course of HGA as a case whose severity requires specific therapeutic interventions/care (e.g., respiratory support, renal replacement therapy, circulatory support, blood transfusion, intensive care monitoring etc.) beyond the administration of antimicrobial therapy. Acute renal injury/renal failure was the most commonly reported complication in monoinfected CRID (Fig 9). For most cases, a prerenal cause was reported. In a few cases, acute kidney injury was attributed to rhabdomyolysis, pigment-induced nephropathy, or nephritis. In most cases, renal function recovered completely after treatment of HGA. For 14 cases with acute kidney injury, temporary hemodialysis was reported. In three cases, the necessity of renal replacement therapy beyond the acute phase of infection was reported (Table 23), but in

two of those cases it was unclear whether renal replacement therapy was permanently necessary.

After multi-organ failure and pneumonitis with/without ARDS, cardiac complications were common. In those cases, it is hard to say whether the cardiac complications were solely attributable to HGA, or which role predisposing factors/underlying comorbidities may have played in their occurrence. In two cases, *A. phagocytophilum* was detected in the heart; in one case with exudative pericardial effusion by PCR in the pericardial fluid [134] and in one fatal case with pancarditis immunohistochemically within cells with mononuclear morphology that had infiltrated the cardiac and epicardial tissues [69]. A small study that looked at cardiac bio-markers of 11 HGA cases found positive markers in five cases, which was associated with a more complicated course of illness [135]. Encephalitis and meningitis were diagnosed in 2.5% and 1.3% of cases, respectively (which is more than in the US-case surveillance from 2008–2012, where encephalitis or meningitis was seen in 0.9% of cases [19]). Other neurological complications affected different parts of the central and peripheral nervous system, mostly presenting as a neuritis and/ or myelitis, in two cases as an acute demyelinating polyradiculoneuritis [77,136]. Most reported CSF examinations were unremarkable. In some cases pleocytosis and/or elevated protein levels in CSF was reported [136–139]. In only one patient with meningitis, bilateral Bell's palsy and lymphocyte-dominant pleocytosis, a positive CSF-PCR and -culture result for HGA was reported [33].

Among the reviewed cases 13 cases (11 monoinfected, 2 with coinfections) presented with hemophagocytic lymphohistiocytosis (HLH). All HLH cases were male and, with the exception of two cases from Asia and one from Europe, all cases were from North America [72,140–149]. Of the two cases with coinfections one had an Influenza B virus infection and one a *B. burgdorferi* as well as a Powassan virus infection [141,149]. The case fatality rate of these HGA associated HLH cases was 23% (3/13), but it has to be acknowledged that two of the fatal cases did not receive appropriate antimicrobial therapy as the diagnosis of anaplasmosis was only made retrospectively, while all other cases were appropriately treated. HLH associated with HGA therefore appears to have a similar mortality compared to infection-associated HLH in general, where case fatality rates of around 20% are reported [150]. In 8 cases, immunosuppressive treatment for HLH was administered (etoposide, corticosteroids, interleukin-1 receptor antagonist, and/or immunoglobulins).

Splenic rupture was reported in 4 cases (all with HGA monoinfection; [151–154]) with a fatal outcome in one of these cases (Table 21; case no. 10). Thus, atraumatic splenic rupture, a complication more commonly reported in babesiosis [154], does not seem to be a common, but a possible complication of HGA.

Secondary opportunistic infections were rare with only 3 cases reported: An 80-year-old, immunocompromised patient (CLL with Richter's transformation, asplenia, high-dose steroid treatment) with multi-organ failure and cryptococcal pneumonia who died on day 5 of his hospitalization (Table 21; case no. 1). A 71-year-old multimorbid patient with fatal pulmonary aspergillosis who died 28 days after start of his symptoms (Table 21; case no. 3). A Chinese patient coinfected with SFTSV who was diagnosed with *Aspergillus* pneumonia as well as tuberculosis [34]. (Note: Infections that are common in patients suffering from a prolonged course of illness with ICU-care, such as *Candida* oesophagitis or HSV1-reactivation, were not classified as opportunistic infections by us).

When comparing the frequency of occurrence of complications in HGA CRID, we found that complications are statistically significantly more common in North America than in Europe and Asia (Table 17). Whether this statistically suggested difference is truly existing, and possibly related to differences in virulence of geographically distinct strains of *A. phagocytophilum*, remains speculative. Also possible is that the differences is explained by the fact that

European patients are on average younger (Table 10). The time between the onset of symptoms and the receipt of appropriate antimicrobial treatment as a possible explanatory factor was shorter in European patients, but did not reach statistical significance (Table 20).

For CRNID, the data on complications was very limited, with two Chinese studies dominating the data. One of the 2 studies reported a rate of severe complications (SIRS and/or MOF) of 50.6% (42/83) and a case fatality rate of 26.5% (22/83), which is significantly higher than in any other cohort study [155]. The other study reported severe complications in 41.2% (26/62) of the cases and a case fatality rate of 8.1–16.7% (depending on the province) [156]. Of note, in both studies some signs and symptoms were reported that have not been reported in context with HGA elsewhere (e.g. relative bradycardia, jaundice and facial edema [155,156]). As the two studies included patients before SFTSV was identified in 2011 and were conducted in a region known to be endemic for SFTSV, it can be speculated that the unusual severity may have resulted from unrecognised SFTSV co-infections [157,158]. This is why some authors suggest that studies on HGA from SFTSV endemic regions should be carefully reviewed if SFTSV was not excluded, especially if symptoms differ from what would normally be expected from HGA [121].

## Treatment

No prospective controlled treatment study has yet been conducted to determine the antimicrobial efficacy in HGA. All patients with a suspected diagnosis of HGA should be treated empirically with appropriate antibiotics without delay, regardless of whether the diagnosis is confirmed, as delayed treatment increases the risk of complications [159]. Doxycycline is the most commonly used antimicrobial agent (Table 18) and the recommended treatment of choice [17]. Our analysis confirms that fever resolves rapidly (mostly within 1 day) after starting appropriate antimicrobial treatment. Thus, in the case fever persists beyond 48h after treatment was started, the diagnosis of HGA should be reconsidered [160].

In our analysis, the median duration of antimicrobial treatment was 14 days, with a clear tendency of physicians to either opt for a 7-, 10-, 14- or 21-day treatment regimens (Fig 10). This is due to the fact that no studies have yet been conducted to define an optimal treatment regimen for HGA. The tendency to opt for 10–14 days is due to the fact that this duration of treatment also covers possible Lyme co-infections. [160]. While tetracycline is contraindicated

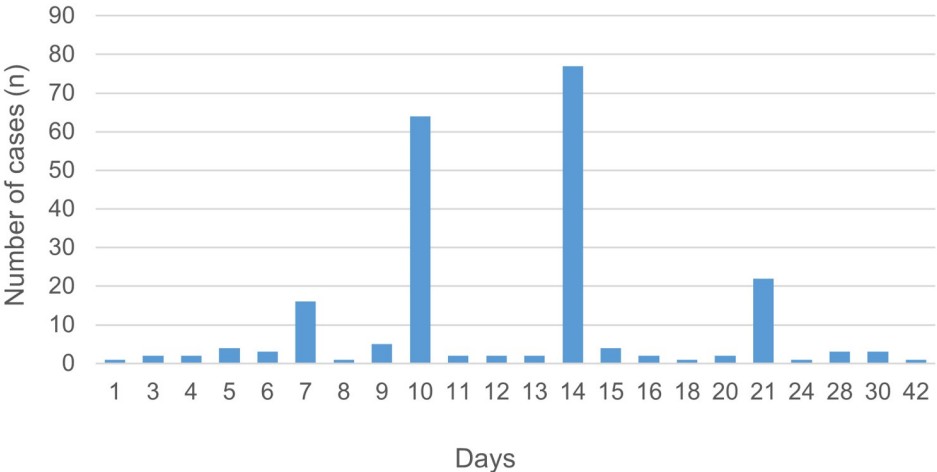

**Fig 10. Duration of antimicrobial treatment for human granulocytotropic anaplasmosis (n = 201).**

in children due to the risk of irreversible dental staining, short administration of doxycycline for less than 22 days is considered safe at any age [161–163], and for children who have no signs of concomitant Lyme disease, the duration of therapy can be shortened to 4–5 days (corresponding to three days after defeverence) [17,160]. The most frequently recommended and most widely adopted dosing regimen for adults is 100mg of doxycycline twice daily. The recommended pediatric dose of doxycycline for children weighing less than 45kg is 2.2mg/kg body weight twice daily [17]. Data on the use of doxycycline in pregnancy is very limited regarding the possible risks of teratogenicity and primary tooth staining. Therefore, the use of doxycycline in pregnancy demands a critical individual risk-benefit assessment [17]. An alternative treatment in pregnancy is rifampicin. In our analysis, 12 cases were treated with rifampicin. Of those, eight were pregnant (Table 13) and two were children (four and six years old), therefore groups where doxycycline was used hesitantly in the past [164]. Two patients received rifampicin and doxycycline consecutively, one of them was initially treated for suspected endocarditis with multiple antibiotic agents including rifampicin and later, after the diagnosis of HGA, switched to doxycycline [165]. The other patient was initially treated with doxycycline for 3 weeks and then switched to rifampicin because PCR testing for *A. phagocytophilum* was persistingly positive [139]. All patients treated with rifampicin recovered. However, even though evidence for in vitro activity of rifampicin against *A. phagocytophilum* exists, clinical trials on the in vivo efficacy of rifampicin in HGA are lacking [166]. Therefore, based on limited case report data, rifampicin could be useful as a therapy for HGA in mild anaplasmosis during pregnancy, or in patients with allergies to tetracyclines [17]. In our analysis, two patients were treated with chloramphenicol. In one patient, chloramphenicol was added to his doxycycline treatment after three days because of persisting symptoms. In the other patient, HGA was primarily treated with chloramphenicol due to known doxycycline intolerance [167,168]. Both patients recovered. However, considering the fact that in vitro testing showed insufficient activity of chloramphenicol against *A. phagocytophilum* [166] chloramphenicol should not be used to treat HGA [17].

Data on the prophylactic administration of a single dose of doxycycline after a tick bite is scarce. The US CDC generally does not recommend prophylactic treatment for asymptomatic individuals due to a recent tick bite. However, there are geographic regions in the U.S. where such prophylaxis for Lyme disease may be recommended under certain circumstances [17,169]. It is unclear, whether such a prophylaxis would be effective for HGA. Of note, our analysis included two cases who suffered from HGA despite receiving such a single dose prophylaxis [170,171].

## Outcome

Most HGA patients recover fully without any sequelae, even patients who do not receive any or inappropriate antimicrobial therapy.

The case fatality rate in our analysis is ~3% in CRID and CRNID, which is higher than the <1% reported by US case surveillance from 2000 to 2012 [4,19]. Most of the reported fatal cases are >50 years of age and suffer from comorbidities. Half of the reported fatal cases did not receive appropriate antimicrobial therapy. Interestingly, no fatal case of a patient with HGA monoinfection has yet been reported from Europe.

In a few cases sequelae are reported (Table 22). All patients with sequelae were appropriately treated with doxycycline. Neurological sequelae were the most common, followed by renal failure requiring dialysis.

Little is known about immunity following infection with *A. phagocytophilum*. Patients infected with *A. phagocytophilum* often develop high antibody titers that may persist for

several years, but whether these antibodies provide protective immunity remains unclear [104]. It is generally assumed that patients are protected from reinfection, at least for a certain period of time, as reports of reinfection are very rare. In the literature, we found only one laboratory-confirmed report of reinfection in a woman who contracted HGA again two years after primary infection [172].

## Limitations

Our analyses have several limitations. Next to publication bias and the retrospective nature of most available data sets, most reviewed studies did only provide incomplete data sets. In addition to the inhomogeneity and incompleteness of the available data sets, the data and results of studies that reported on case series or cohorts were often given as total values, medians or percentages, so that it was often not possible to assign the data to individual cases (see section on CRNID). Even though we tried to eliminate case duplicates, the cohort studies we looked at could potentially contain duplicate cases already described in other cohort studies or case reports. When we compare our findings in case studies with the cohorts we looked at as well as CDC data on HGA in the USA, we see that the course of illness in our cases was comparatively severe, which points to reporting and publication bias as mentioned above. Since the currently available clinical data is limited to regions with the respective diagnostic capacities (Table 4) extrapolating to other regions is difficult. Therefore, our study provides an overview of how HGA presents in these regions, whereas for large parts of the world we do not know whether HGA exists there and, if so, whether its clinical presentation is uniform. Another limitation of our study is that we did not include all languages. Box 1 summarizes the main conclusions we have drawn from our review and Box 2 contains our selection of publications that we recommend clinicians to read.

### Box 1

**Key learning points:**

- The presence of *A. phagocytophilum* and human cases of granulocytotropic anaplasmosis (HGA) are reported from all continents except from Antarctica

- HGA usually presents as a non-specific febrile illness accompanied by thrombocytopenia, leukopenia and elevated liver function tests

- Although usually mild and self-limiting, HGA may cause severe and even life-threatening complications (e.g. acute renal failure, ARDS, and multi organ failure in 9.8%, 6.3%, and 7.5% of the cases)

- The antimicrobial treatment of choice is with doxycycline and treatment response is usually fast with fever subsiding within 1 day after starting treatment

- Overall, sequelae following HGA are rare and lethality of HGA is low

- Unlike in human monocytotropic ehrlichiosis (HME), reports of opportunistic infections complicating HGA are rare

- HGA during pregnancy does not appear to be associated with unfavorable outcomes

Box 2

**Key references:**

1. Biggs HM, Behravesh CB, Bradley KK, Dahlgren FS, Drexler NA, Dumler JS, et al. Diagnosis and Management of Tickborne Rickettsial Diseases: Rocky Mountain Spotted Fever and Other Spotted Fever Group Rickettsioses, Ehrlichioses, and Anaplasmosis—United States. MMWR Recomm Rep. 2016;65(2):1–44. Epub 2016/05/14. doi: 10.15585/mmwr.rr6502a1. PubMed PMID: 27172113.

2. Bakken JS, Dumler JS. Human granulocytic anaplasmosis. Infect Dis Clin North Am. 2015;29(2):341–55. Epub 2015/05/23. doi: 10.1016/j.idc.2015.02.007. PubMed PMID: 25999228; PubMed Central PMCID: PMCPMC4441757.

3. Wang F, Yan M, Liu A, Chen T, Luo L, Li L, et al. The seroprevalence of *Anaplasma phagocytophilum* in global human populations: A systematic review and meta-analysis. Transbound Emerg Dis. 2020. Epub 2020/03/18. doi: 10.1111/tbed.13548. PubMed PMID: 32180352.

4. Dahlgren FS, Mandel EJ, Krebs JW, Massung RF, McQuiston JH. Increasing incidence of *Ehrlichia chaffeensis* and *Anaplasma phagocytophilum* in the United States, 2000–2007. Am J Trop Med Hyg. 2011;85(1):124–31. Epub 2011/07/08. doi: 10.4269/ajtmh.2011.10-0613. PubMed PMID: 21734137; PubMed Central PMCID: PMCPMC3122356.

5. Dahlgren FS, Heitman KN, Drexler NA, Massung RF, Behravesh CB. Human granulocytic anaplasmosis in the United States from 2008 to 2012: a summary of national surveillance data. Am J Trop Med Hyg. 2015;93(1):66–72. Epub 2015/04/15. doi: 10.4269/ajtmh.15-0122. PubMed PMID: 25870428; PubMed Central PMCID: PMCPMC4497906.

6. Matei IA, Estrada-Peña A, Cutler SJ, Vayssier-Taussat M, Varela-Castro L, Potkonjak A, et al. A review on the eco-epidemiology and clinical management of human granulocytic anaplasmosis and its agent in Europe. Parasit Vectors. 2019;12(1):599. Epub 2019/12/23. doi: 10.1186/s13071-019-3852-6. PubMed PMID: 31864403; PubMed Central PMCID: PMCPMC6925858.

## Supporting information

**S1 Text. Systematic review protocol.**
(DOCX)

**S2 Text. Search terms used.**
(DOCX)

**S3 Text. Reference list of considered publications.**
(DOCX)

**S4 Text. PRISMA checklist.**
(DOCX)

**S5 Text. Additional performed analyses of CRID.**
(DOCX)

**S6 Text. Additional analyses of CRNID.**
(DOCX)

**S1 Table. Data extraction sheet.**
(DOCX)

**S2 Table. Laboratory reference values used.**
(DOCX)

**S3 Table. Master table of raw data.**
(XLSX)

## Author Contributions

**Conceptualization:** Esther Kuenzli, Andreas Neumayr.

**Data curation:** Sophie Schudel, Larissa Gygax.

**Formal analysis:** Sophie Schudel, Christian Kositz.

**Methodology:** Christian Kositz, Esther Kuenzli, Andreas Neumayr.

**Resources:** Larissa Gygax.

**Supervision:** Christian Kositz, Esther Kuenzli, Andreas Neumayr.

**Validation:** Larissa Gygax.

**Visualization:** Sophie Schudel.

**Writing – original draft:** Sophie Schudel.

**Writing – review & editing:** Christian Kositz, Esther Kuenzli, Andreas Neumayr.

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
