## [Decision Letter · Decision Letter 0]

1 Apr 2024

Dear Dr. Kositz,

Thank you very much for submitting your manuscript "Human granulocytotropic anaplasmosis – a systematic review and analysis of the literature" for consideration at PLOS Neglected Tropical Diseases. As with all papers reviewed by the journal, your manuscript was reviewed by members of the editorial board and by several independent reviewers. In light of the reviews (below this email), we would like to invite the resubmission of a significantly-revised version that takes into account the reviewers' comments. 

We cannot make any decision about publication until we have seen the revised manuscript and your response to the reviewers' comments. Your revised manuscript is also likely to be sent to reviewers for further evaluation.

Sincerely,

Job E Lopez, Ph.D.

Academic Editor

Ana LTO Nascimento

Section Editor

Reviewer's Responses to Questions

**Key Review Criteria Required for Acceptance?**

**Methods**

-Are the objectives of the study clearly articulated with a clear testable hypothesis stated?

-Is the study design appropriate to address the stated objectives?

-Is the population clearly described and appropriate for the hypothesis being tested?

-Is the sample size sufficient to ensure adequate power to address the hypothesis being tested?

-Were correct statistical analysis used to support conclusions?

-Are there concerns about ethical or regulatory requirements being met?

Reviewer #1: The objectives of this study are clearly stated.The required design for performing a systematic review were followed.There are no concerns about ethical or regulatory requirements.

Reviewer #2: Not hypothesis-driven research. The study design is appropriate to address the stated objectives. The publication selection methods is clearly described and largely appropriate. The sample size used is adequate for a small systematic review and this is reflected by the PRISMA checklist. Summary statistics only are used. No ethical or regulatory concerns. Specific items:

Page 7, line 94. The citation regarding incidence of HGA in the US covered only 208-2012. The peak of HGA cases in the US was until recently 2017, and now 2021. Why not update this with the data that is readily available on the CDC website?

Page 7, line 96. Spelling error, should be “tick-borne”

Page 7, line 99. H. concinna has also been implicated as a potential vector in China.

Page 8, Table 1. Same comment as above

Page 8, HGA has been identified associated with organ transplantation (please see Transpl Infect Dis. 2019 Aug;21(4):e13129. doi: 10.1111/tid.13129)

Page 8, line 120. Most existing data suggest that rash is rare and when present often reflects erythema migrans after B. burgdorferi coinfection (which is also infrequent).

Page 8, line 122. Similarly, although reported, a critical evaluation does not often support a clear diagnosis of meningoencephalitis because of lack of objective findings for that diagnosis.

Page 9, I was confused about why HME was being discussed, so consider elaborating about the purpose of the introductory comments. I didn’t understand this until I read the entire intro.

Page 9, line 129. While this had been true in the US, it has not been widely described this way for quite awhile, and this is reflected in the nomenclature used by the CDC, and in major texts and online resources (e.g. UpToDate).

Page 9, lines 138-139. True but in fact it is recommended directly by clinical suspicion alone.

Page 9, lines 148-149. I don't understand this comment. The data on the 25-75% of patients for whom a blood smear will reveal morulae is from immunocompetent patients. Most now believe that rate is closer to 90-100% with an adequate blood smear evaluation. Thus, how could this be insensitive, and how would this justify the conclusion of a low bacterial burden, this is a relatively HIGH blood bacterial burden disease which is why PCR alone works so well.

Page 11, Table 2. Column 3 (Disadvantages), Row “Serology”. Cross reactions with Rickettsia spp. is actually quite infrequent, it would be better to state cross reactivity with other Anaplasmataceae since such cross reactions with other rickettsial diseases is rare.

**Results**

-Does the analysis presented match the analysis plan?

-Are the results clearly and completely presented?

-Are the figures (Tables, Images) of sufficient quality for clarity?

Reviewer #1: The analysis does match the analysis plan.The results are clearly and completely presented.The figures are satisfactory.

There are numerous aspects which should be addressed or corrected:

1. The meaning of "paraclinical" is unclear.

2. The inappropriate term, liver function tests, is inaccurate. The mentioned tests are "liver injury tests".

3. HME was not reviewed systematically, and comparisons of HGA and HME are unnecessary and inappropriate in a systematic review.

4. Table 2: What is meant by serum " scar"?

5. Please provide a reference for the statement about serology : "decreased sensitivity after early administration of appropriate antibiotics".

6. Awarding a certainty grade B for IFA or ELISA antibody titer of 64 is highly questionable.

7. Table 7: Reported co-infections are questionable, especially Bartonella , Rickettsia, Ehrlichia chaffeensis, Coxiella , Orientia, dengue virus, and chikungunya virus. How well documented are these as active co-infection versus seroprevalence?

8. Table 8: How does microscopy distinguish between Ehrlichia ewingi and Anaplasma phagocytophilum, especially in states where Anaplasma is undetected?

9. Neologisms such as immunocompromization should be corrected to immunocompromised.

10. Clinical signs such as Bell's palsy and arthralgia are more likely due to a diagnosed or undiagnosed co-infection.

11. Lines 198: Please cite figure 8 not figure 3.

12 Figures 9 and 10 do not show "number of cases" as depicted in the y-axis.

13. Figure 9: What are the criteria used for diagnoses of cardiomyopathy, heart failure, and disseminated intravascular ccoagulation which seem to be very unlikely caused by Anaplasma.

14. Lines 563-4:There are no antibiotics listed in Table 19.

15. Line 632: The cases are primarily in the northeast not the northwest.

16. Line 638: Are there any detections of Anaplasma phagocytophilum in vectors in South and Central America?

17. Line 648: "Insensitive" is not the correct concept; "inadequate" would be more appropriate.

18. Line 671: Was the patient infected in Brazil or French Guiana?

19. Line 741: Please correct the spelling of Anaplasma phagocytophilum.

20. Lines of 788-790: Actually Lyme disease is currently spreading southward into eastern Tennessee and Western North Carolina.

Reviewer #2: The analysis matched the plan. The results are generally clearly and completely presented, and the figures of good quality. Specific issues include:

Page 13, line 217. “immunosuppression” please define what this includes since the CDC reporting includes an array of predisposing conditions not classically felt to be immune compromising in the way that HIV, organ transplantation or immunosuppressive therapies are.

Page 13, lines 225-229. I am guessing that many of the individual case reports provided results from >1 approach named here. How were those classified? As is evident a grade A+ would be appropriate alone when another test was also done, but the lack of serconversion in such patients would argue for a possible false positive A+ method result. Additionally, those cases with >1 diagnostic from the categories of A and B+, might also be considered equivalent to A certainty. How these are handled is critical since the final analyses used should rely solely on the highest certainty testing, including that for coinfections, which as far as I can tell was not addressed. 

Page 13, line 230. CDC case definition of HGA. Please note that these diagnostic criteria are for surveillance purposes and not necessarily recommended for diagnosis. There is still ongoing debate about the application of diagnostics that should be recognized.

Page 13, line 231. “IgM IFA” Please note that the CDC does NOT advocate the use of IgM serology for the diagnosis of HGA. Including this here might signify that this is an acceptable serodiagnostic approach when it is not.

Page 14, Table 3, Column 2 (Description), Row 7 (Serology – IgG). I personally do not agree with this application, although I know it appears on the CDC surveillance website; however, even there the CDC admits that these are surveillance criteria, not diagnostic. The issue here is best addressed in this publication: doi: 10.1128/JCM.40.7.2612-2615.2002

Page 23, Table 7. To evaluate co-infections, this is a potentially misleading table since no grading system to asses certainty of a correct diagnosis was used for the coinfections. In my experience, this is a very significant problem that leads to the misimpression that co-infections are common. This document should strive to provide such data objectively rather than accepting the coinfection diagnoses in published literature. Additionally, the TBEV row is a bit confounded by the fact that TBEV does NOT occur in North America; thus using the denominator calculated by summing the numbers of all globally reported cases provides a misleading result and interpretation. Also, the B. burgdorferi s.l. + Bartonella henselae row is particularly troubling since B henselae is NOT tick-transmitted and there are NO CRID for this combination?? And finally, for the Ehrlichia chaffeensis row, is this even plausible? 

Page 25, line 390. Do the analyses reflect the inclusion of ANY diagnostic criteria group or only the highest quality (A+)? Please describe this as relevant for each analysis, but preferably, select a single diagnostic group that provides high confidence in the diagnosis and use that for analysis. Anything less would introduce significant potentially troubling analyses and would lack confidence in the results and conclusions made.

Page 26, Table 10, hospitalization. It is interesting that the hospitalization rate is about twice as high in North America in these papers vs. what is accrued in case reports for the CDC. This indicates the bias of this approach, which is acknowledged in the Discussion. Please consider more emphasis on this bias in the abstract and conclusions.

Page 27, lines 416-417. “vertical transmission” - you might clarify what this means, since it demonstrates the capacity for transplacental transmission in humans.

Page 28, Table 11, patient no. 9, culprit blood produce – minor error, “RCB” should be “RBC”; also patient 11 had leukoreduced RBCs. I know and it’s in the paper.

Page 30, Table 12, patient 17, treatment category nomenclature. I did not review this manuscript, but if this is a quote, please use quotation marks; if not, please simply describe the antimicrobial treatment without judgement regarding "appropriateness", and consider the same throughout the manuscript.

Page 39, lines 56-564. The substances considered "appropriate" are not listed in Table 19 as was indicated.

Page 39, Table 18. Clearly this cannot be the entire list of antimicrobial agents used? Is this a representation of those drugs considered "appropriate" or effective?

Page 45, Table 22. Patient no. 7, Complications column. “renal replacement therapy” What does this mean? Is this kidney transplant? If so, say that; if not known, please use an alternative phrase to indicate the renal therapy was not defined/identified.

**Conclusions**

-Are the conclusions supported by the data presented?

-Are the limitations of analysis clearly described?

-Do the authors discuss how these data can be helpful to advance our understanding of the topic under study?

-Is public health relevance addressed?

Reviewer #1: The interpretation of the data and accurate statements about the limitations and reservations about the interpretations are quite appropriate.

Reviewer #2: Conclusions are generally well-supported by the data, and limitations are discussed, but need some additional consideration especially with regard to the discrepancy between the overall conclusions discerned from major series publications and those that use CDC national data. This likely is the result of publication bias and although discussed, I think this needs considerably more emphasis. Also, the use of ALL data rather than that based on a diagnostic quality grading may be including data from cases erroneously classified as anaplasmosis. I would recommend an analysis of those cases that use only A or A+ diagnostic grade data and compare to the overall set or the set that is not A or A+ grade. The same situation is relevant for coinfections, which seem considerable higher than anticipated based on reputable publications that have directly sought this information, and is likely driven by publication bias for coinfections, some of which that were included that are highly implausible and questionable, for which NONE of the coinfections was graded for quality. Additional individual concerns are:

Page 47, line 632. Places where majority of cases are reported in North America. First, this location refers to the U.S not North America, and most cases are reported from the Northeast and Upper Midwest, not the Northwest.

Page 47, line 638. About the cases from South America, especially Venezuela….for the authors awareness, I worked closely with the Venezuelan group considering this in platelets and found NO evidence of infection by any molecular method we used. The obviously found a way to publish without molecular data, but likely did not acknowledge the extensive negative data that had already been done. This is a reason to focus on the data within the trustworthy diagnostic group.

Page 49, lines 676-678. From my reading of this, there is no evidence that this Candidatus species even has the capacity to cause disease - the main topic of this systematic review.

Page 49, lines 680-681. in addition to the appropriate commentary about this, one must consider addressing the climate in the US and some European countries regarding the wide-spread belief among some activists that co-infections are exceedingly common (>75-100% of infections diagnosed as some boutique "Lyme-literate" physician offices and laboratories advocate for this and often cosponsor misleading publications that appear to be valid. The 16% coinfection rate here is likely a significant overestimate as most studies directly examining this in a systematic and unbiased fashion find <10%.

Page 49, lines 687-690. Another key consideration that should be address

---

## [Editor Report · Decision Letter 1]

11 Jun 2024

Dear DR. Kositz,

Thank you very much for submitting your manuscript "Human granulocytotropic anaplasmosis – a systematic review and analysis of the literature" for consideration at PLOS Neglected Tropical Diseases. As with all papers reviewed by the journal, your manuscript was reviewed by members of the editorial board and by several independent reviewers. In light of the reviews (below this email), we would like to invite the resubmission of a significantly-revised version that takes into account the reviewers' comments. 

We cannot make any decision about publication until we have seen the revised manuscript and your response to the reviewers' comments. Your revised manuscript is also likely to be sent to reviewers for further evaluation.

Sincerely,

Ana LTO Nascimento

Section Editor

Ana LTO Nascimento

Section Editor
---

## [Editor Report · Decision Letter 2]

21 Jun 2024

Dear Dr. Kositz,

We are pleased to inform you that your manuscript 'Human granulocytotropic anaplasmosis – a systematic review and analysis of the literature' has been provisionally accepted for publication in PLOS Neglected Tropical Diseases.

Best regards,

Ana LTO Nascimento

Section Editor

Ana LTO Nascimento

Section Editor

---

## [Editor Report · Acceptance letter]

29 Jul 2024

Dear Dr. Kositz,

We are delighted to inform you that your manuscript, "Human granulocytotropic anaplasmosis – a systematic review and analysis of the literature," has been formally accepted for publication in PLOS Neglected Tropical Diseases.

Best regards,

Shaden Kamhawi

co-Editor-in-Chief

Paul Brindley

co-Editor-in-Chief
